# PLANE: Representation Learning over Planar Graphs

**Radoslav Dimitrov**[*][†]
Department of Computer Science
University of Oxford
contact@radoslav11.com

**Zeyang Zhao**[*][†]
Department of Computer Science
University of Oxford
zeyzang.zhao@cs.ox.ac.uk

**Ralph Abboud**[*]
Department of Computer Science
University of Oxford
ralph@ralphabb.ai

**İsmail İlkan Ceylan**
Department of Computer Science
University of Oxford
ismail.ceylan@cs.ox.ac.uk

## Abstract

Graph neural networks iteratively compute representations of nodes of an input graph through a series of transformations in such a way that the learned graph function is isomorphism invariant on graphs, which makes the learned representations *graph invariants*. On the other hand, it is well-known that graph invariants learned by these class of models are *incomplete*: there are pairs of non-isomorphic graphs which cannot be distinguished by standard graph neural networks. This is unsurprising given the computational difficulty of graph isomorphism testing on general graphs, but the situation begs to differ for special graph classes, for which efficient graph isomorphism testing algorithms are known, such as planar graphs. The goal of this work is to design architectures for *efficiently* learning *complete* invariants of planar graphs. Inspired by the classical planar graph isomorphism algorithm of Hopcroft and Tarjan, we propose PLANE as a framework for planar representation learning. PLANE includes architectures which can learn *complete* invariants over planar graphs while remaining practically scalable. We validate the strong performance of PLANE architectures on various planar graph benchmarks.

## 1 Introduction

Graphs are used for representing relational data in a wide range of domains, including physical [53], chemical [17, 36], and biological [21, 68] systems, which led to increasing interest in machine learning (ML) over graphs. Graph neural networks (GNNs) [23, 51] have become prominent for graph ML for a wide range of tasks, owing to their capacity to explicitly encode desirable relational inductive biases [6]. GNNs iteratively compute representations of nodes of an input graph through a series of transformations in such a way that the learned graph-level function represents a *graph invariant*: a property of graphs which is preserved under all isomorphic transformations.

Learning functions on graphs is challenging for various reasons, particularly since the learning problem contains the infamous graph isomorphism problem, for which the best known algorithm, given in a breakthrough result by Babai [4], runs in quasi-polynomial time. A large class of GNNs can thus only learn *incomplete* graph invariants for scalablity purposes. In fact, standard GNNs are known to be at most as expressive as the 1-dimensional Weisfeiler-Leman algorithm (1-WL)[59] in terms of distinguishing power [45, 63]. There are simple non-isomorphic pairs of graphs, such as the pair shown in Figure 1, which cannot be distinguished by 1-WL and by a large class of GNNs. This

---

[*]This work is largely conducted while these authors were still affiliated with the University of Oxford.
[†]Equal contribution.

limitation motivated a large body of work aiming to explain and overcome the expressiveness barrier of these architectures [1, 5, 7, 9, 11, 15, 37, 42, 43, 45, 50].

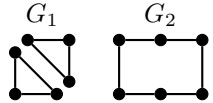

The expressiveness limitations of standard GNNs already apply on planar graphs, since, e.g., the graphs $G_1$ and $G_2$ from Figure 1 are planar. There are, however, efficient and complete graph isomorphism testing algorithms for planar graphs [28, 29, 41], which motivates an aligned design of dedicated architectures with better properties over planar graphs. Building on this idea, we propose architectures for *efficiently* learning *complete* invariants over planar graphs.

Figure 1: Two graphs indistinguishable by 1-WL.

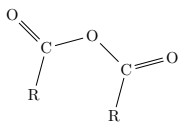

Figure 2: Acid anhydride as a planar graph.

Planar structures appear, e.g., in road networks [62], circuit design [8], and most importantly, in biochemistry [54]. Molecules are commonly encoded as planar graphs with node and edge types, as shown in Figure 2, which enables the application of many graph ML architectures on biochemistry tasks. For example, *all* molecule datasets in OGBG [33] consist of planar graphs. Biochemistry has been a very important application domain for both classical graph isomorphism testing[3] and for graph ML. By designing architectures for learning *complete* invariants over planar graphs, our work unlocks the full potential of both these domains.

The contributions of this work can be summarized as follows:

- Building on the classical literature, we introduce PLANE as a framework for learning *isomorphism-complete* invariant functions over planar graphs and derive the BASEPLANE architecture (Section 5). We prove that BASEPLANE is a *scalable*, *complete* learning algorithm on planar graphs: BASEPLANE can distinguish *any* pair of non-isomorphic planar graphs (Section 6).

- We conduct an empirical analysis for BASEPLANE evaluating its *expressive power* on three tasks (Section 7.1), and validating its *performance* on real-world graph classification (Section 7.2) and regression benchmarks (Section 7.3 and 7.4). The empirical results support the presented theory, and also yield multiple state-of-the-art results for BASEPLANE on molecular datasets.

The proofs and additional experimental details are delegated to the appendix of this paper.

## 2   A primer on graphs, invariants, and graph neural networks

**Graphs.** Consider simple, undirected graphs $G = (V, E, \zeta)$, where $V$ is a set of nodes, $E \subseteq V \times V$ is a set of edges, and $\zeta : V \to \mathbb{C}$ is a (coloring) function. If the range of this map is $\mathbb{C} = \mathbb{R}^d$, we refer to it as a $d$-dimensional *feature map*. A graph is *connected* if there is a path between any two nodes and disconnected otherwise. A graph is *biconnected* if it cannot become disconnected by removing any single node. A graph is *triconnected* if the graph cannot become disconnected by removing any two nodes. A graph is *planar* if it can be drawn on a plane such that no two edges intersect.

**Components.** A *biconnected component* of a graph $G$ is a maximal biconnected subgraph. Any connected graph $G$ can be efficiently decomposed into a tree of biconnected components called the *Block-Cut tree* of $G$ [30], which we denote as BLOCKCUT$(G)$. The blocks are attached to each other at shared nodes called *cut nodes*. A *triconnected component* of a graph $G$ is a maximal triconnected subgraph. Triconnected components of a graph $G$ can also be compiled (very efficiently [27]) into a tree, known as the *SPQR tree* [16], which we denote as SPQR$(G)$. Given a graph $G$, we denote by $\sigma^G$ the set of all SPQR components of $G$ (i.e., nodes of SPQR$(G)$), and by $\pi^G$ the set of all biconnected components of $G$. Moreover, we denote by $\sigma_u^G$ the set of all SPQR components of $G$ where $u$ appears as a node, and by $\pi_u^G$ the set of all biconnected components of $G$ where $u$ appears as a node.

**Labeled trees.** We sometimes refer to rooted, undirected, labeled trees $\Gamma = (V, E, \zeta)$, where the canonical root node is given as one of tree's *centroids*: a node with the property that none of its branches contains more than half of the other nodes. We denote by ROOT$(\Gamma)$ the canonical root of $\Gamma$, and define the *depth* $d_u$ of a node $u$ in the tree as the node's minimal distance from the canonical root. The *children* of a node $u$ is the set $\chi(u) = \{v | (u, v) \in E, d_v = d_u + 1\}$. The *descendants* of a node

---

[3]Graph isomorphism first appears in the chemical documentation literature [49], as the problem of matching a molecular graph against a database of such graphs; see, e.g., Grohe and Schweitzer [26] for a recent survey.

$u$ is given by set of all nodes reachable from $u$ through a path of length $k \geq 0$ such that the node at position $j + 1$ has one more depth than the node at position $j$, for every $0 \leq j \leq k$. Given a rooted tree $\Gamma$ and a node $u$, the *subtree* of $\Gamma$ rooted at node $u$ is the tree induced by the descendants of $u$, which we donote by $\Gamma_u$. For technical convenience, we allow the induced subtree $\Gamma_u$ of a node $u$ even if $u$ does not appear in the tree $\Gamma$, in which case $\Gamma_u$ is the empty tree.

**Node and graph invariants.** An *isomorphism* from a graph $G = (V, E, \zeta)$ to a graph $G' = (V', E', \zeta')$ is a bijection $f : V \rightarrow V'$ such that $\zeta(u) = \zeta'(f(u))$ for all $u \in V$, and $(u, v) \in E$ if and only if $(f(u), f(v)) \in E'$, for all $u, v \in V$. A *node invariant* is a function $\xi$ that associates with each graph $G = (V, E, \zeta)$ a function $\xi(G)$ defined on $V$ such that for all graphs $G$ and $G'$, all isomorphisms $f$ from $G$ to $G'$, and all nodes $u \in V$, it holds that $\xi(G)(u) = \xi(G')(f(u))$. A *graph invariant* is a function $\xi$ defined on graphs such that $\xi(G) = \xi(G')$ for all isomorphic graphs $G$ and $G'$. We can derive a graph invariant $\xi$ from a node invariant $\xi'$ by mapping each graph $G$ to the multiset $\{\!\{\xi'(G)(u) \mid u \in V\}\!\}$. We say that a graph invariant $\xi$ *distinguishes* two graphs $G$ and $G'$ if $\xi(G) \neq \xi(G')$. If a graph invariant $\xi$ distinguishes $G$ and $G'$ then there is no isomorphism from $G$ to $G'$. If the converse also holds, then $\xi$ is a *complete* graph invariant. We can speak of (in)completeness of invariants on special classes of graphs, e.g., 1-WL computes an incomplete invariant on general graphs, but it is well-known to compute a complete invariant on trees [24].

**Message passing neural networks.** A vast majority of GNNs are instances of *message passing neural networks (MPNNs)* [22]. Given an input graph $G = (V, E, \zeta)$, an MPNN sets, for each node $u \in V$, an initial node representation $\zeta(u) = \boldsymbol{h}_u^{(0)}$, and iteratively computes representations $\boldsymbol{h}_u^{(\ell)}$ for a fixed number of layers $0 \leq \ell \leq L$ as:

$$\boldsymbol{h}_u^{(\ell+1)} := \phi\left(\boldsymbol{h}_u^{(\ell)}, \psi(\boldsymbol{h}_u^{(\ell)}, \{\!\{\boldsymbol{h}_v^{(\ell)} \mid v \in N_u\}\!\})\right),$$

where $\phi$ and $\psi$ are respectively *update* and *aggregation* functions, and $N_u$ is the neighborhood of $u$. Node representations can be *pooled* to obtain graph-level embeddings by, e.g., summing all node embeddings. We denote by $\boldsymbol{z}^{(L)}$ the resulting graph-level embeddings. In this case, an MPNN can be viewed as an encoder that maps each graph $G$ to a representation $\boldsymbol{z}_G^{(L)}$, computing a graph invariant.

## 3 Related work

The expressive power of MPNNs is upper bounded by 1-WL [45, 63] in terms of distinguishing graphs, and by the logic $\mathsf{C}^2$ in terms of capturing functions over graphs [5], motivating a body of work to improve on these bounds. One notable direction has been to enrich node features with unique node identifiers [42, 65], random discrete colors [15], or random noisy dimensions [1, 50]. Another line of work proposes *higher-order* architectures [37, 43–45] based on higher-order tensors [44], or a higher-order form of message passing [45], which typically align with a $k$-dimensional WL test[4], for some $k > 1$. Higher-order architectures are not scalable, and most existing models are upper bounded by 2-WL (or, *oblivious* 3-WL). Another body of work is based on sub-graph sampling [7, 9, 11, 56], with pre-set sub-graphs used within model computations. These approaches can yield substantial expressiveness improvements, but they rely on manual sub-graph selection, and require running expensive pre-computations. Finally, MPNNs have been extended to incorporate other graph kernels, i.e., shortest paths [2, 64], random walks [46, 47] and nested color refinement [67].

The bottleneck limiting the expressiveness of MPNNs is the implicit need to perform graph isomorphism checking, which is challenging in the general case. However, there are classes of graphs, such as planar graphs, with efficient and complete isomorphism algorithms, thus eliminating this bottleneck. For planar graphs, the first complete algorithm for isomorphism testing was presented by Hopcroft and Tarjan [29], and was followed up by a series of algorithms [10, 19, 28, 52]. Kukluk et al. [41] presented an algorithm, which we refer to as KHC, that is more suitable for practical applications, and that we align with in our work. As a result of this alignment, our approach is the first *efficient* and *complete* learning algorithm on planar graphs. Observe that 3-WL (or, *oblivious* 4-WL) is also known to be complete on planar graphs [38], but this algorithm is far from being scalable [35] and as a result there is no neural, learnable version implemented. By contrast, our architecture learns representations of efficiently computed components and uses these to obtain refined representations. Our approach extends the inductive biases of MPNNs based on structures, such as biconnected components, which are recently noted to be beneficial in the literature [66].

---

[4]We write $k$-WL to refer to the folklore (more standard) version of the algorithm following Grohe [25].

## 4 A practical planar isomorphism algorithm

The idea behind the KHC algorithm is to compute a canonical code for planar graphs, allowing us to reduce the problem of isomorphism testing to checking whether the codes of the respective graphs are equal. Importantly, we do not use the codes generated by KHC in our model, and view codes as an abstraction through which alignment with KHC is later proven. Formally, we can define a code as a string over the alphabet $\Sigma \cup \mathbb{N}$: for each graph, the KHC algorithm computes codes for various components resulting from decompositions, and gradually builds a code representation for the graph. For readability, we allow reserved symbols "(", ")" and "," in the generated codes. We present an overview of KHC and refer to Kukluk et al. [41] for details. We can assume that the planar graphs are connected, as the algorithm can be extended to disconnected graphs by independently computing the codes for each of the components, and then concatenating them in their lexicographical order.

**Generating a code for the the graph.** Given a connected planar graph $G = (V, E, \zeta)$, KHC decomposes $G$ into a Block-Cut tree $\delta = \text{BLOCKCUT}(G)$. Every node $u \in \delta$ is either a cut node of $G$, or a virtual node associated with a biconnected component of $G$. KHC iteratively removes the leaf nodes of $\delta$ as follows: if $u$ is a leaf node associated with a biconnected component $B$, KHC uses a subprocedure BICODE to compute the canonical code $\text{CODE}(\delta_u) = \text{BICODE}(B)$ and removes $u$ from the tree; otherwise, if the leaf node $u$ is a cut node, KHC overrides the initial code $\text{CODE}((\{u\}, \{\}))$, using an aggregate code of the removed biconnected components in which $u$ occurs as a node. This procedure continues until there is a single node left, and the code for this remaining node is taken as the code of the entire connected graph. This process yields a complete graph invariant. Conceptually, it is more convenient for our purposes to reformulate this procedure as follows: we first canonically root $\delta$, and then code the subtrees in a *bottom-up procedure*. Specifically, we iteratively generate codes for subtrees and the final code for $G$ is then simply the code of $\delta_{\text{ROOT}(\delta)}$.

**Generating a code for biconnected components.** KHC relies on a subprocedure BICODE to compute a code, given a biconnected planar graph $B$. Specifically, it uses the SPQR tree $\gamma = \text{SPQR}(B)$ which uniquely decomposes a biconnected component into a tree with virtual nodes of one of four types: $S$ for *cycle* graphs, $P$ for *two-node dipole* graphs, $Q$ for a graph that has a *single edge*, and finally $R$ for a triconnected component that is *not* a dipole or a cycle. We first generate codes for the induced sub-graphs based on these virtual nodes. Then the SPQR tree is canonically rooted, and similarly to the procedure on the Block-Cut tree, we iteratively build codes for the subtrees of $\gamma$ in a bottom-up fashion. Due to the simpler structure of the SPQR tree, instead of making overrides, the recursive code generation is done by prepending a number $\theta(C, C')$ for a parent SPQR tree node $C$ and each $C' \in \chi(C)$. This number is generated based on the way $C$ and $C'$ are connected in $B$. Generating codes for the virtual $P$ and $Q$ nodes is trivial. For $S$ nodes, we use the lexicographically smallest ordering of the cycle, and concatenate the individual node codes. However, for $R$ nodes we require a more complex procedure and this is done using Weinberg's algorithm as a subroutine.

**Generating a code for triconnected components.** Whitney [60] has shown that triconnected graphs have only two planar embeddings, and Weinberg introduced an algorithm that computes a canonical code for triconnected planar graphs [58] which we call TRICODE. This code is used as one of the building blocks of the KHC algorithm and can be extended to labeled triconnected planar graphs, which is essential for our purposes. Weinberg's algorithm [58] generates a canonical order for a triconnected component $T$, by traversing all the edges in both directions via a walk. Let $\omega$ be the sequence of visited nodes in this particular walk and write $\omega[i]$ to denote $i$-th node in it. This walk is then used to generate a sequence $\kappa$ of same length, that corresponds to the order in which we first visit the nodes: for each node $u = \omega[i]$ that occurs in the walk, we set $\kappa[i] = 1 + |\{\kappa[j] \mid j < i\}|$ if $\omega[i]$ is the first occurrence of $u$, or $\kappa[i] = \kappa[\min\{j \mid \omega[i] = \omega[j]\}]$ otherwise. For example, the walk $\omega = \langle v_1, v_3, v_2, v_3, v_1 \rangle$ yields $\kappa = \langle 1, 2, 3, 2, 1 \rangle$. Given such a walk of length $k$ and a corresponding sequence $\kappa$, we compute the following canonical code: $\text{TRICODE}(T) = (\kappa[1], \text{CODE}((\{\omega[1]\}, \{\}))), \ldots, (\kappa[k], \text{CODE}((\{\omega[k]\}, \{\})))$.

## 5 PLANE: Representation learning over planar graphs

KHC generates a unique code for every planar graph in a hierarchical manner based on the decompositions. Our framework aligns with this algorithm: given an input graph $G = (V, E, \zeta)$, PLANE first computes the BLOCKCUT and SPQR trees (**A – C**), and then learns representations for the nodes, the components, and the whole graph (**D – G**), as illustrated in Figure 3.

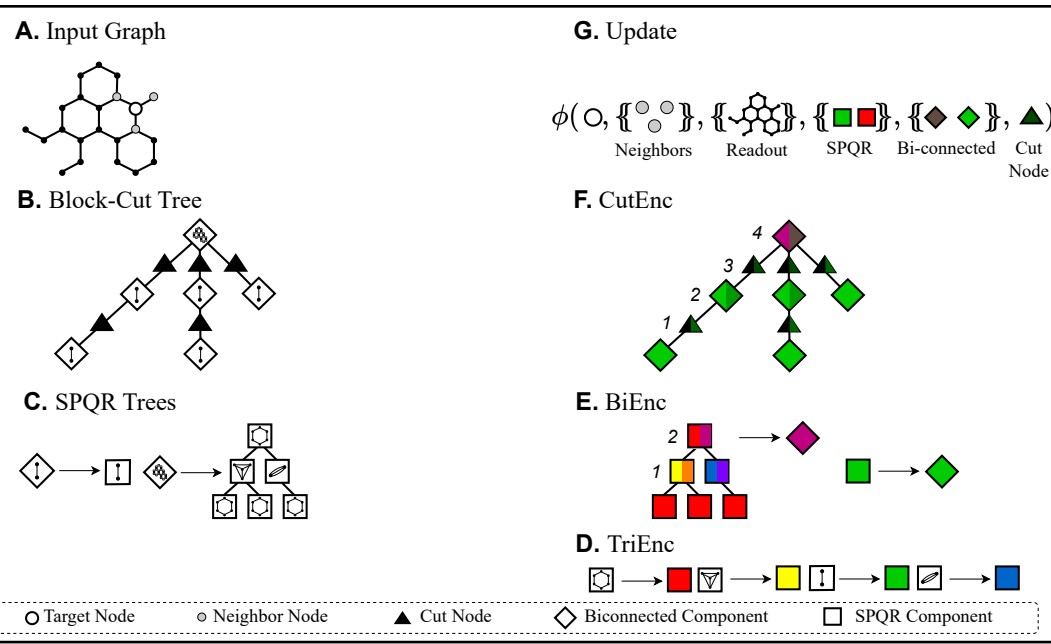

Figure 3: Given an input graph (A) PlanE computes the BLOCKCUT (B) and SPQR (C) trees and assigns initial embeddings to SPQR components using TRIENC (D). Then, BIENC iteratively traverses each SPQR tree bottom-up (numbered), updates SPQR embeddings, and computes biconnected component embeddings (E). CUTENC then analogously traverses the BLOCKCUT tree to compute a graph embedding (F). Finally, the node representation is updated (G).

Formally, PLANE sets the initial node representations $\boldsymbol{h}_u^{(0)} = \zeta(u)$ for each node $u \in V$, and computes, for every layer $1 \leq \ell \leq L$, the representations $\widehat{\boldsymbol{h}}_C^{(\ell)}$ of SPQR components of $\sigma^G$, the representations $\widetilde{\boldsymbol{h}}_B^{(\ell)}$ of biconnected components $B \in \pi^G$, the representations $\overline{\boldsymbol{h}}_{\delta_u}^{(\ell)}$ of subtrees in the Block-Cut tree $\delta = \text{BLOCKCUT}(G)$ for each node $u \in V$, and the representations $\boldsymbol{h}_u^{(\ell)}$ of nodes as:

$$\widehat{\boldsymbol{h}}_C^{(\ell)} = \text{TRIENC}\big(C, \{(\boldsymbol{h}_u^{(\ell-1)}, u)|\ u \in C\}\big)$$

$$\widetilde{\boldsymbol{h}}_B^{(\ell)} = \text{BIENC}\big(B, \{(\widehat{\boldsymbol{h}}_C^{(\ell)}, C)|\ C \in \sigma^B\}\big)$$

$$\overline{\boldsymbol{h}}_{\delta_u}^{(\ell)} = \text{CUTENC}\big(\delta_u, \{(\boldsymbol{h}_v^{(\ell-1)}, v)|\ v \in V\}, \{(\widetilde{\boldsymbol{h}}_B^{(\ell)}, B)|\ B \in \pi^G\}\big)$$

$$\boldsymbol{h}_u^{(\ell)} = \phi\big(\boldsymbol{h}_u^{(\ell-1)}, \{\!\!\{\boldsymbol{h}_v^{(\ell-1)}|\ v \in N_u\}\!\!\}, \{\!\!\{\boldsymbol{h}_v^{(\ell-1)}|\ v \in V\}\!\!\}, \{\!\!\{\widehat{\boldsymbol{h}}_T^{(\ell)}|\ T \in \sigma_u^G\}\!\!\}, \{\!\!\{\widetilde{\boldsymbol{h}}_B^{(\ell)}|\ B \in \pi_u^G\}\!\!\}, \overline{\boldsymbol{h}}_{\delta_u}^{(\ell)}\big)$$

where TRIENC, BIENC, and CUTENC are invariant encoders and $\phi$ is an UPDATE function. The encoders TRIENC and BIENC are analogous to TRICODE and BICODE of the KHC algorithm. In PLANE, we further simplify the final graph code generation, by learning embeddings for the cut nodes, which is implemented by the CUTENC encoder. For graph-level tasks, we apply a *pooling* function on final node embeddings to obtain a graph-level embedding $\boldsymbol{z}_G^{(L)}$.

There are many choices for deriving PLANE architectures, but we propose a simple model, BASE-PLANE, to clearly identify the virtue of the model architecture which aligns with KHC, as follows:

**TRIENC.** Given an SPQR component $C$ and the node representations $\boldsymbol{h}_u^{(\ell-1)}$ of each node $u$, TRIENC encodes $C$ based on the walk $\omega$ given by Weinberg's algorithm, and its corresponding sequence $\kappa$ as:

$$\widehat{\boldsymbol{h}}_C^{(\ell)} = \text{MLP}\left(\sum_{i=1}^{|\omega|} \text{MLP}\left(\boldsymbol{h}_{\omega[i]}^{(\ell-1)} \| \boldsymbol{p}_{\kappa[i]} \| \boldsymbol{p}_i\right)\right),$$

where $\boldsymbol{p}_x \in \mathbb{R}^d$ is the positional embedding [57]. This is a simple sequence model with a positional encoding on the walk, and a second one based on the generated sequence $\kappa$. Edge features can also be

concatenated while respecting the edge order given by the walk. The nodes of $\textsc{Spqr}(G)$ are one of the types $S, P, Q, R$, where for $S, P, Q$ types, we have a trivial ordering for the induced components, and Weinberg's algorithm also gives an ordering for $R$ nodes that correspond to triconnected components.

**BiEnc.** Given a biconnected component $B$ and the representations $\boldsymbol{h}_C^{(\ell)}$ of each component induced by a node $C$ in $\gamma = \textsc{Spqr}(B)$, BiEnc uses the SPQR tree and the integers $\theta(C, C')$ corresponding to how we connect $C$ and $C' \in \chi(C)$. BiEnc then computes a representation for each subtree $\gamma_C$ induced by a node $C$ in a bottom up fashion as:

$$\widetilde{\boldsymbol{h}}_{\gamma_C}^{(\ell)} = \mathrm{MLP}\left(\widehat{\boldsymbol{h}}_C^{(\ell)} + \sum_{C' \in \chi(C)} \mathrm{MLP}\left(\widetilde{\boldsymbol{h}}_{\gamma_{C'}}^{(\ell)} \| \boldsymbol{p}_{\theta(C,C')}\right)\right).$$

This encoder operates in a bottom up fashion to ensure that a subtree representation of the children of $C$ exists before it encodes the subtree $\gamma_C$. The representation of the canonical root node in $\gamma$ is used as the representation of the biconnected component $B$ by setting: $\widetilde{\boldsymbol{h}}_B^{(\ell)} = \widetilde{\boldsymbol{h}}_{\gamma_{\textsc{root}(\gamma)}}^{(\ell)}$.

**CutEnc.** Given a subtree $\delta_u$ of a Block-Cut tree $\delta$, the representations $\widetilde{\boldsymbol{h}}_B^{(\ell)}$ of each biconnected component $B$, and the node representations $\boldsymbol{h}_u^{(\ell-1)}$ of each node, CutEnc sets $\overline{\boldsymbol{h}}_{\delta_u}^{(\ell)} = \boldsymbol{0}^{d(\ell)}$ if $u$ is *not* a cut node; otherwise, it computes the subtree representations as:

$$\overline{\boldsymbol{h}}_{\delta_u}^{(\ell)} = \mathrm{MLP}\left(\boldsymbol{h}_u^{(\ell-1)} + \sum_{B \in \chi(u)} \mathrm{MLP}\left(\widetilde{\boldsymbol{h}}_B^{(\ell)} + \sum_{v \in \chi(B)} \overline{\boldsymbol{h}}_{\delta_v}^{(\ell)}\right)\right).$$

The CutEnc procedure is called in a bottom-up order to ensure that the representations of the grandchildren are already computed. We learn the cut node subtree representations instead of employing the hierarchical overrides that are present in the KHC algorithm, as the latter is not ideal in a learning algorithm. However, with sufficient layers, these representations are complete invariants.

**Update.** Putting these altogether, we update the node representations $\boldsymbol{h}_u^{(\ell)}$ of each node $u$ as:

$$\boldsymbol{h}_u^{(\ell)} = f^{(\ell)}\Big(g_1^{(\ell)}\big(\boldsymbol{h}_u^{(\ell-1)} + \sum_{v \in N_u} \boldsymbol{h}_v^{(\ell-1)}\big) \| g_2^{(\ell)}\big(\sum_{v \in V} \boldsymbol{h}_v^{(\ell-1)}\big) \|$$
$$g_3^{(\ell)}\big(\boldsymbol{h}_u^{(\ell-1)} + \sum_{C \in \sigma_u^G} \widehat{\boldsymbol{h}}_C^{(\ell)}\big) \| g_4^{(\ell)}\big(\boldsymbol{h}_u^{(\ell-1)} + \sum_{B \in \pi_u^G} \widetilde{\boldsymbol{h}}_B^{(\ell)}\big) \| \overline{\boldsymbol{h}}_{\delta_u}^{(\ell)}\Big),$$

where $f^{(\ell)}$ and $g_i^{(\ell)}$ are either linear maps or two-layer MLPs. Finally, we pool as:

$$\boldsymbol{z}_G = \mathrm{MLP}\left(\|_{\ell=1}^L \Big(\sum_{u \in V^G} \boldsymbol{h}_u^{(\ell)}\Big)\right).$$

## 6 Expressive power and efficiency of BASEPLANE

We present the theoretical result of this paper, which states that BASEPLANE can distinguish any pair of planar graphs, even when using only a logarithmic number of layers in the size of the input graphs:

**Theorem 6.1.** *For any planar graphs $G_1 = (V_1, E_1, \zeta_1)$ and $G_2 = (V_2, E_2, \zeta_2)$, there exists a parametrization of BASEPLANE with at most $L = \lceil \log_2(\max\{|V_1|, |V_2|\}) \rceil + 1$ layers, which computes a complete graph invariant, that is, the final graph-level embeddings satisfy $\boldsymbol{z}_{G_1}^{(L)} \neq \boldsymbol{z}_{G_2}^{(L)}$ if and only if $G_1$ and $G_2$ are not isomorphic.*

The construction is non-uniform, since the number of layers needed depends on the size of the input graphs. In this respect, our result is similar to other results aligning GNNs with 1-WL with sufficiently many layers [45, 63]. There are, however, two key differences: (i) BASEPLANE computes isomorphism-complete invariants over planar graphs and (ii) our construction requires only logarithmic number of layers in the size of the input graphs (as opposed to linear).

The theorem builds on the properties of each encoder being complete. We first show that a single application of TRIENC and BIENC is sufficient to encode all relevant components of an input graph in an isomorphism-complete way:

**Lemma 6.2.** *Let $G = (V, E, \zeta)$ be a planar graph. Then, for any biconnected components $B, B'$ of $G$, and for any SPQR components $C$ and $C'$ of $G$, there exists a parametrization of the functions* TRIENC *and* BIENC *such that:*

(i) $\widetilde{\boldsymbol{h}}_B \neq \widetilde{\boldsymbol{h}}_{B'}$ *if and only if $B$ and $B'$ are not isomorphic, and*

(ii) $\widehat{\boldsymbol{h}}_C \neq \widehat{\boldsymbol{h}}_{C'}$ *if and only if $C$ and $C'$ are not isomorphic.*

Intuitively, this result follows from a natural alignment to the procedures of the KHC algorithm: the existence of unique codes for different components is proven for the algorithm and we lift this result to the embeddings of the respective graphs, using the universality of MLPs [14, 31, 32].

Our main result rests on a key result related to CUTENC, which states that BASEPLANE computes complete graph invariants for all subtrees of the Block-Cut tree. We use an inductive proof, where the logarithmic bound stems from a single layer computing complete invariants for all subtrees induced by cut nodes that have at most one grandchild cut node, the induced subtree of which is incomplete.

**Lemma 6.3.** *For a planar graph $G = (V, E, \zeta)$ of order $n$ and its associated Block-Cut tree $\delta = \text{BLOCKCUT}(G)$, there exists a $L = \lceil \log_2(n) \rceil + 1$ layer parametrization of* BASEPLANE *that computes a complete graph invariant for each subtree $\delta_u$ induced by each cut node $u$.*

With Lemma 6.2 and Lemma 6.3 in place, Theorem 6.1 follows from the fact that every biconnected component and every cut node of the graph are encoded in an isomorphism-complete way, which is sufficient for distinguishing planar graphs.

**Runtime efficiency.** Theoretically, the runtime of one BasePlanE layer is $\mathcal{O}(|V|d^2)$ with a (one-off) pre-processing time $\mathcal{O}(|V|^2)$ as we elaborate in Appendix B. In practical terms, this makes BasePlanE linear in the number of graph nodes after preprocessing. This is very scalable as opposed to other complete algorithms based on 3-WL (or, oblivious 4-WL).

# 7 Experimental evaluation

In this section, we evaluate BASEPLANE in three different settings. First, we conduct three experiments to evaluate the expressive power of BASEPLANE. Second, we evaluate a BASEPLANE variant using edge features on the MolHIV graph classification task from OGB [33, 61]. Finally, we evaluate this variant on graph regression over ZINC [18] and QM9 [12, 48]. We provide an additional experiment on the runtime of BASEPLANE in Appendix B, and an ablation study on ZINC in Appendix C. All experimental details, including hyperparameters, can be found in Appendix D[5].

## 7.1 Expressiveness evaluation

### 7.1.1 Planar satisfiability benchmark: EXP

**Experimental setup.** We evaluate BASEPLANE on the planar EXP benchmark [1] and compare with standard MPNNs, MPNNs with random node initialization and (higher-order) 3-GCNs [45]. EXP consists of planar graphs which each represent a satisfiability problem (SAT) instance. These instances are grouped into pairs, such that these pairs cannot be distinguished by 1-WL, but lead to different SAT outcomes. The task in EXP is to predict the satisfiability of each instance. To obtain above-random performance on this dataset, a model must have a sufficiently strong expressive power (2-WL or more). To conduct this experiment, we use a 2-layer BASE-PLANE model with 64-dimensional node embeddings. We instantiate the triconnected component encoder with 16-dimensional positional encodings, each computed using a periodicity of 64.

Table 1: Accuracy results on EXP. Baselines are from Abboud et al. [1].

| Model | Accuracy (%) |
|---|---|
| GCN | $50.0_{\pm 0.00}$ |
| GCN-RNI(N) | $98.0_{\pm 1.85}$ |
| 3-GCN | $\mathbf{99.7}_{\pm 0.004}$ |
| BASEPLANE | $\mathbf{100}_{\pm 0.00}$ |

---

[5]Across all experiments, we present tabular results, and follow a convention in which the **best** result is bold in **black**, and the second best result is shown in bold in gray.

**Results.** All results are reported in Table 1. As expected, BASEPLANE perfectly solves the task, achieving a performance of 100% (despite not relying on any higher-order method). BASEPLANE solely relies on classical algorithm component decompositions, and does not rely on explicitly selected and designed features, to achieve this performance gain. This experiment highlights that the general algorithmic decomposition effectively improves expressiveness in a practical setup, and leads to strong performance on EXP, where a standard MPNN would otherwise fail.

### 7.1.2 Planar 3-regular graphs: P3R

**Experimental setup.** We propose a new synthetic dataset P3R based on 3-regular planar graphs, and experiment with BASEPLANE, GIN and 2-WL-expressive PPGN [43]. For this experiment, we generated all 3-regular planar graphs of size 10, leading to exactly 9 non-isomorphic graphs. For each such graph, we generated 50 isomorphic graphs by permuting their nodes. The task is then to predict the correct class of an input graph, where the random accuracy is approximately $11.1\%$. This task is challenging given the regularity of the graphs.

Table 2: Accuracy results on P3R.

| Model | Accuracy (%) |
| --- | --- |
| GIN | 11.1 $_{\pm 0.00}$ |
| PPGN | **100** $_{\pm 0.00}$ |
| BASEPLANE | **100** $_{\pm 0.00}$ |

**Results.** We report all results in Table 2. As expected, GIN struggles to go beyond a random guess, whereas BASEPLANE and PPGNs easily solve the task, achieving 100% accuracy.

### 7.1.3 Clustering coefficients of QM9 graphs: QM9$_{CC}$

In this experiment, we evaluate the ability of BASEPLANE to detect structural graph signals *without* an explicit reference to the target structure. To this end, we propose a simple, yet challenging, synthetic task: given a subset of graphs from QM9, we aim to predict the graph-level *clustering coefficient (CC)*. Computing CC requires counting triangles in the graph, which is impossible to solve with standard MPNNs [65].

**Data.** We select a subset QM9$_{CC}$ of graphs from QM9 to obtain a diverse distribution of CCs. As most QM9 graphs have a CC of 0, we consider graphs with a CC in the interval $[0.06, 0.16]$, as this range has high variability. We then normalize the CCs to the unit interval $[0, 1]$. We apply the earlier filtering on the original QM9 splits to obtain train/validation/test sets that are direct subsets of the full QM9 splits, and which consist of 44226, 3941 and 3921 graphs, respectively.

**Experimental setup.** Given the small size of QM9 and the locality of triangle counting, we use 32-dimensional node embeddings and 3 layers across all models. Moreover, we use a common 100 epoch training setup for fairness. For evaluation, we report mean absolute error (MAE) on the test set, averaged across 5 runs. For this experiment, our baselines are (i) an input-agnostic constant prediction that returns a minimal test MAE, (ii) the MPNNs GCNs [40] and GIN [63], (iii) ESAN [7], an MPNN that computes sub-structures through node and edge removals, but which does *not* explicitly extract triangles, and (iv) BASEPLANE, using 16-dimensional positional encoding vectors.

**Results.** Results on QM9$_{CC}$ are provided in Table 3. BASEPLANE comfortably outperforms standard MPNNs. Indeed, GCN performance is only marginally better than the constant baseline and GIN's MAE is over an order of magnitude behind BASEPLANE. This is a very substantial gap, and confirms that MPNNs are unable to accurately detect triangles to compute CCs. Moreover, BASEPLANE achieves an MAE over 40% lower than ESAN. Overall, BASEPLANE effectively detects triangle structures, despite these not being explicitly provided, and thus its underlying algorithmic decomposition effectively captures latent structural graph properties in this setting.

Table 3: MAE of BASEPLANE and baselines on the QM9$_{CC}$ dataset.

| Model | MAE |
| --- | --- |
| Constant | 0.1627 $_{\pm 0.0000}$ |
| GCN | 0.1275 $_{\pm 0.0012}$ |
| GIN | 0.0612 $_{\pm 0.0018}$ |
| ESAN | 0.0038 $_{\pm 0.0010}$ |
| BASEPLANE | **0.0023** $_{\pm 0.0004}$ |

**Performance analysis.** To better understand our results, we visualize the predictions of BASEPLANE, GIN, and GCN using scatter plots in Figure 4. As expected, BASEPLANE follows the ideal regression line. By contrast, GIN and GCN are much less stable. Indeed, GIN struggles with CCs at the extremes of the $[0, 1]$ range, but is better at intermediate values, whereas GCN is consistently unreliable, and rarely returns predictions above 0.7. This highlights the structural limitation of GCNs, namely its self-loop mechanism for representation updates, which causes ambiguity for detecting triangles.

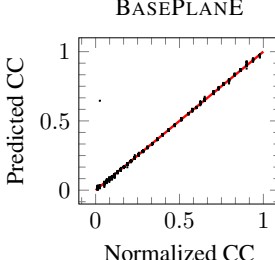 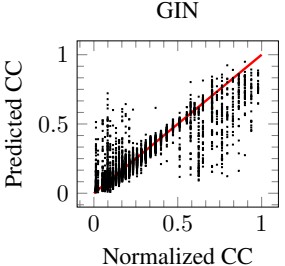 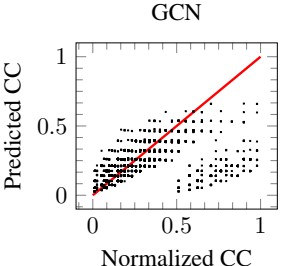

Figure 4: Scatter plots of BASEPLANE, GIN, and GCN predictions versus the true normalized CC. The red line represents ideal behavior where predictions match the true normalized CC.

## 7.2 Graph classification on MolHIV

**Model setup.** We use a BASEPLANE variant that uses edge features, called E-BASEPLANE (defined in the appendix), on OGB [33] MolHIV and compare against baselines. We instantiate E-BASEPLANE with an embedding dimension of 64, 16-dimensional positional encodings, and report the average ROC-AUC across 10 independent runs.

**Results.** The results for E-BASEPLANE and other baselines on MolHIV are shown in Table 4: Despite not explicitly extracting relevant cycles and/or molecular sub-structures, E-BASEPLANE outperforms standard MPNNs and the domain-specific HIMP model. It is also competitive with substructure-aware models CIN and GSN, which include dedicated structures for inference. Therefore, E-BASEPLANE performs strongly in practice with minimal design effort, and effectively uses its structural inductive bias to remain competitive with dedicated architectures.

Table 4: ROC-AUC of BASE-PLANE and baselines on MolHIV.

| | |
|---|---|
| GCN [40] | $75.58_{\pm 0.97}$ |
| GIN [63] | $77.07_{\pm 1.40}$ |
| PNA [13] | $79.05_{\pm 1.32}$ |
| ESAN [7] | $78.00_{\pm 1.42}$ |
| GSN [11] | $80.39_{\pm 0.90}$ |
| CIN [9] | $\mathbf{80.94_{\pm 0.57}}$ |
| HIMP [20] | $78.80_{\pm 0.82}$ |
| E-BASEPLANE | $80.04_{\pm 0.50}$ |

## 7.3 Graph regression on QM9

**Experimental setup.** We map QM9 [48] edge types into features by defining a learnable embedding per edge type, and subsequently apply E-BASEPLANE to the dataset. We evaluate E-BASEPLANE on all 13 QM9 properties following the same splits and protocol (with MAE results averaged over 5 test set reruns) of GNN-FiLM [12]. We compare R-SPN against GNN-FiLM models and their fully adjacent (FA) variants [3], as well as shortest path networks (SPNs) [2]. We report results with an 3-layer E-BASEPLANE using 128-dimensional node embeddings and 32-dimensional positional encodings.

**Results.** E-BASEPLANE results on QM9 are provided in Table 5. In this table, E-BASEPLANE outperforms high-hop SPNs, despite being simpler and more efficient, achieving state-of-the-art results on 9 of the 13 tasks. The gains are particularly prominent on the first five properties, where R-SPNs originally provided relatively little improvement over FA models, suggesting that E-BASEPLANE offers complementary structural advantages to SPNs. This was corroborated in our experimental tuning: E-BASEPLANE performance peaks around 3 layers, whereas SPN performance continues to improve up to 8 (and potentially more) layers, which suggests that E-BASEPLANE is more efficient at directly communicating information, making further message passing redundant.

Overall, E-BASEPLANE maintains the performance levels of R-SPN with a smaller computational footprint. Indeed, messages for component representations efficiently propagate over trees in E-BASEPLANE, and the number of added components is small (see appendix for more details). Therefore E-BASEPLANE and the PLANE framework offer a more scalable alternative to explicit higher-hop neighborhood message passing over planar graphs.

Table 5: MAE of E-BASEPLANE and baselines on QM9. Other model results and their fully adjacent (FA) extensions are as previously reported [2, 3].

| Property | R-GIN | | R-GAT | | R-SPN | | BASEPLANE |
| | base | +FA | base | +FA | $k = 5$ | $k = 10$ | |
|---|---|---|---|---|---|---|---|
| mu | $2.64_{\pm 0.11}$ | $2.54_{\pm 0.09}$ | $2.68_{\pm 0.11}$ | $2.73_{\pm 0.07}$ | $2.16_{\pm 0.08}$ | $2.21_{\pm 0.21}$ | $\mathbf{1.97}_{\pm 0.03}$ |
| alpha | $4.67_{\pm 0.52}$ | $2.28_{\pm 0.04}$ | $4.65_{\pm 0.44}$ | $2.32_{\pm 0.16}$ | $1.74_{\pm 0.05}$ | $1.66_{\pm 0.06}$ | $\mathbf{1.63}_{\pm 0.01}$ |
| HOMO | $1.42_{\pm 0.01}$ | $1.26_{\pm 0.02}$ | $1.48_{\pm 0.03}$ | $1.43_{\pm 0.02}$ | $1.19_{\pm 0.04}$ | $1.20_{\pm 0.08}$ | $\mathbf{1.15}_{\pm 0.01}$ |
| LUMO | $1.50_{\pm 0.09}$ | $1.34_{\pm 0.04}$ | $1.53_{\pm 0.07}$ | $1.41_{\pm 0.03}$ | $1.13_{\pm 0.01}$ | $1.20_{\pm 0.06}$ | $\mathbf{1.06}_{\pm 0.02}$ |
| gap | $2.27_{\pm 0.09}$ | $1.96_{\pm 0.04}$ | $2.31_{\pm 0.06}$ | $2.08_{\pm 0.05}$ | $1.76_{\pm 0.03}$ | $1.77_{\pm 0.06}$ | $\mathbf{1.73}_{\pm 0.02}$ |
| R2 | $15.63_{\pm 1.40}$ | $12.61_{\pm 0.37}$ | $52.39_{\pm 42.5}$ | $15.76_{\pm 1.17}$ | $10.59_{\pm 0.35}$ | $10.63_{\pm 1.01}$ | $\mathbf{10.53}_{\pm 0.55}$ |
| ZPVE | $12.93_{\pm 1.81}$ | $5.03_{\pm 0.36}$ | $14.87_{\pm 2.88}$ | $5.98_{\pm 0.43}$ | $3.16_{\pm 0.06}$ | $\mathbf{2.58}_{\pm 0.13}$ | $2.81_{\pm 0.16}$ |
| U0 | $5.88_{\pm 1.01}$ | $2.21_{\pm 0.12}$ | $7.61_{\pm 0.46}$ | $2.19_{\pm 0.25}$ | $1.10_{\pm 0.03}$ | $\mathbf{0.89}_{\pm 0.05}$ | $0.95_{\pm 0.04}$ |
| U | $18.71_{\pm 23.36}$ | $2.32_{\pm 0.18}$ | $6.86_{\pm 0.53}$ | $2.11_{\pm 0.10}$ | $1.09_{\pm 0.05}$ | $\mathbf{0.93}_{\pm 0.03}$ | $0.94_{\pm 0.04}$ |
| H | $5.62_{\pm 0.81}$ | $2.26_{\pm 0.19}$ | $7.64_{\pm 0.92}$ | $2.27_{\pm 0.29}$ | $1.10_{\pm 0.03}$ | $\mathbf{0.92}_{\pm 0.03}$ | $\mathbf{0.92}_{\pm 0.04}$ |
| G | $5.38_{\pm 0.75}$ | $2.04_{\pm 0.24}$ | $6.54_{\pm 0.36}$ | $2.07_{\pm 0.07}$ | $1.04_{\pm 0.04}$ | $\mathbf{0.83}_{\pm 0.05}$ | $0.88_{\pm 0.04}$ |
| Cv | $3.53_{\pm 0.37}$ | $1.86_{\pm 0.03}$ | $4.11_{\pm 0.27}$ | $2.03_{\pm 0.14}$ | $1.34_{\pm 0.03}$ | $1.23_{\pm 0.06}$ | $\mathbf{1.20}_{\pm 0.06}$ |
| Omega | $1.05_{\pm 0.11}$ | $0.80_{\pm 0.04}$ | $1.48_{\pm 0.87}$ | $0.73_{\pm 0.04}$ | $0.53_{\pm 0.02}$ | $0.52_{\pm 0.02}$ | $\mathbf{0.45}_{\pm 0.01}$ |

## 7.4 Graph regression on ZINC

**Experimental setup.** We (i) evaluate BASEPLANE on the ZINC subset (12k graphs) without edge features, (ii) evaluate E-BASEPLANE on this subset and on the full ZINC dataset (500k graphs). To this end, we run BASEPLANE and E-BASEPLANE with 64 and 128-dimensional embeddings, 16-dimensional positional embeddings, and 3 layers. For evaluation, we compute MAE on the respective test sets, and report the best average of 10 runs across all experiments.

**Results.** Results on ZINC are shown in Table 6: Both BASEPLANE and E-BASEPLANE perform strongly, with E-BASEPLANE achieving state-of-the-art performance on ZINC12k with edge features and both models outperforming all but one baseline in the other two settings. These results are very promising, and highlight the robustness of (E-)BASEPLANE.

Table 6: MAE of (E-)BASEPLANE and baselines on ZINC.

| Edge Features | ZINC(12k) No | ZINC(12k) Yes | ZINC(Full) Yes |
|---|---|---|---|
| GCN [40] | $0.278_{\pm 0.003}$ | - | - |
| GIN(-E) [34, 63] | $0.387_{\pm 0.015}$ | $0.252_{\pm 0.014}$ | $0.088_{\pm 0.002}$ |
| PNA [13] | $0.320_{\pm 0.032}$ | $0.188_{\pm 0.004}$ | $0.320_{\pm 0.032}$ |
| GSN [11] | $0.140_{\pm 0.006}$ | $0.101_{\pm 0.010}$ | - |
| CIN [9] | $\mathbf{0.115}_{\pm 0.003}$ | $0.079_{\pm 0.006}$ | $\mathbf{0.022}_{\pm 0.002}$ |
| ESAN [7] | - | $0.102_{\pm 0.003}$ | - |
| HIMP [20] | - | $0.151_{\pm 0.006}$ | $0.036_{\pm 0.002}$ |
| (E-)BASEPLANE | $0.124_{\pm 0.004}$ | $\mathbf{0.076}_{\pm 0.003}$ | $0.028_{\pm 0.002}$ |

## 8 Limitations, discussions, and outlook

Overall, both BASEPLANE and E-BASEPLANE perform strongly across all our experimental evaluation tasks, despite competing against specialized models in each setting. Moreover, both models are isomorphism-complete over planar graphs. This implies that these models benefit substantially from the structural inductive bias and expressiveness of classical planar algorithms, which in turn makes them a reliable, efficient, and robust solution for representation learning over planar graphs.

Though the PLANE framework is a highly effective and easy to use solution for planar graph representation learning, it is currently limited to planar graphs. Indeed, the classical algorithms underpinning PLANE do not naturally extend beyond the planar graph setting, which in turn limits the applicability of the approach. Thus, a very important avenue for future work is to explore alternative (potentially incomplete) graph decompositions that strike a balance between structural inductive bias, efficiency and expressiveness on more general classes of graphs.

## Acknowledgments and Disclosure of Funding

The authors would like to thank the anonymous reviewers for their feedback which led to substantial improvements in the presentation of the paper. The authors would like to also acknowledge the use of the University of Oxford Advanced Research Computing (ARC) facility in carrying out this work (http://dx.doi.org/10.5281/zenodo.22558).

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

# A Proofs of the statements

In this section, we provide the proofs of the statements from the main paper. Throughout the proofs, we make the standard assumption that the initial node features are from a compact space $K \subseteq \mathbb{R}^d$, for some $d \in \mathbb{N}^+$. We also often need to canonically map elements of finite sets to the integer domain: given a finite set $S$, and any element $x \in S$ of this set, the function $\Psi(S, x) : S \to \{1, \ldots, |S|\}$ maps $x$ to a unique integer index given by some fixed order over the set $S$. Furthermore, we also often need to injectively map sets into real numbers, which is given by the following lemma.

**Lemma A.1.** *Given a finite set $S$, there exists an injective map $g : \mathbb{P}(S) \to [0, 1]$.*

*Proof.* For each subset $M \subseteq S$, consider the mapping:

$$g(M) = \frac{1}{1 + \sum_{x \in M} \Psi(S, x)|S|^{\Psi(M,x)}},$$

which clearly satisfies $g(M_1) \neq g(M_2)$ for any $M_1 \neq M_2 \subseteq S$. $\qquad\square$

We now provide proofs for Lemma 6.2, Lemma 6.3 which are essential for Theorem 6.1. Let us first prove Lemma 6.2:

**Lemma 6.2.** *Let $G = (V, E, \zeta)$ be a planar graph. Then, for any biconnected components $B, B'$ of $G$, and for any SPQR components $C$ and $C'$ of $G$, there exists a parameterization of the functions* TRIENC *and* BIENC *such that:*

  *(i) $\widetilde{\boldsymbol{h}}_B \neq \widetilde{\boldsymbol{h}}_{B'}$ if and only if $B$ and $B'$ are not isomorphic, and*

  *(ii) $\widehat{\boldsymbol{h}}_C \neq \widehat{\boldsymbol{h}}_{C'}$ if and only if $C$ and $C'$ are not isomorphic.*

*Proof.* We show this by first giving a parameterization of TRIENC to distinguish all SPQR components, and then use this construction to give a parameterization of BIENC to distinguish all biconnected components. The proof aligns the encoders with the code generation procedure in the KHC algorithm. The parameterization needs only a single layer, so we will drop the superscripts in the node representations and write, e.g., $\boldsymbol{h}_u$, for brevity. The construction yields single-dimensional real-valued vectors, and, for notational convenience, we view the resulting representations as reals.

**Parameterizing TRIENC.** We initialize the node features as $\boldsymbol{h}_u = \zeta(u)$, where $\zeta : V \to \mathbb{R}^d$. Given an SPQR component $C$ and the initial node representations $\boldsymbol{h}_u$ of each node $u$, TRIENC encodes $C$ based on the walk $\omega$ given by Weinberg's algorithm, and its corresponding sequence $\kappa$ as:

$$\widehat{\boldsymbol{h}}_C = \text{MLP} \left( \sum_{i=1}^{|\omega|} \text{MLP} \left( \boldsymbol{h}_{\omega[i]} \| \boldsymbol{p}_{\kappa[i]} \| \boldsymbol{p}_i \right) \right). \tag{1}$$

In other words, we rely on the walks $\omega$ generated by Weinberg's algorithm: we create a walk on the triconnected components that visits each edge exactly twice. Let us fix $n = 4(|V| + |E| + 1)$, which serves as an upper bound for size of the walks $\omega$, and the size of the walk-induced sequence $\kappa$.

We show a bijection between the multiset of codes $\mathcal{F}$ (given by the KHC algorithm), and the multiset of representations $\mathcal{M}$ (given by TRIENC), where the respective multisets are defined, based on $G$, as follows:

  - $\mathcal{F} = \{\!\!\{ \text{CODE}(C) \mid C \in \sigma^G \}\!\!\}$: the multiset of all codes of the SPQR components of $G$.

  - $\mathcal{M} = \{\!\!\{ \widehat{\boldsymbol{h}}_C \mid C \in \sigma^G \}\!\!\}$: the multiset of the representations of the SPQR components of $G$.

Specifically, we prove the following:

**Claim 1.** *There exists a bijection $\rho$ between $\mathcal{F}$ and $\mathcal{M}$ such that, for any SPQR component $C$:*

$$\rho(\widehat{\boldsymbol{h}}_C) = \text{CODE}(C) \text{ and } \rho^{-1}(\text{CODE}(C)) = \widehat{\boldsymbol{h}}_C.$$

Once Claim 1 established, the desired result is immediate, since, using the bijection $\rho$, and the completeness of the codes generated by the KHC algorithm, we obtain:

$$\widehat{\boldsymbol{h}}_C \neq \widehat{\boldsymbol{h}}_{C'}$$
$$\Leftrightarrow$$
$$\rho^{-1}(\text{CODE}(C)) \neq \rho^{-1}(\text{CODE}(C'))$$
$$\Leftrightarrow$$
$$\text{CODE}(C) \neq \text{CODE}(C')$$
$$\Leftrightarrow$$

*C and $C'$ are non-isomorphic.*

To prove the Claim 1, first note that the canonical code given by Weinberg's algorithm for any component $C$ has the form:

$$\text{CODE}(C) = \text{TRICODE}(C) = (\kappa[1], \text{CODE}((\{\omega[1]\}, \{\}))), \ldots, (\kappa[k], \text{CODE}((\{\omega[k]\}, \{\})))$$
$$= (\kappa[1], \zeta(\omega[1])), \ldots, (\kappa[k], \zeta(\omega[k])). \tag{2}$$

There is a trivial bijection between the codes of the form (2) and sets of the following form:

$$S_C = \{(\zeta(\omega[i]), \kappa[i], i) \mid 1 \leq i \leq |\omega|\}, \tag{3}$$

and, as a result, for each component $C$, we can refer to the set $S_C$ that represents this component $C$ instead of $\text{CODE}(C)$. Sets of this form are of bounded size (since the number of walks are bounded in $C$) and each of their elements is from a countable set. By Lemma A.1 there exists an injective map $g$ between such sets and the interval $[0, 1]$. Since the size of every such set is bounded, and every tuple is from a countable set, we can apply Lemma 5 of Xu et al. [63], and decompose the function $g$ as:

$$g(S_C) = \phi\left(\sum_{x \in S_C} f(x)\right),$$

for an appropriate choice of $\phi : \mathbb{R}^{d'} \to [0, 1]$ and $f(x) \in \mathbb{R}^{d'}$. Based on the structure of the elements of $S_C$, we can further rewrite this as follows:

$$g(S_C) = \phi\left(\sum_{i=1}^{|\omega|} f\left((\zeta(\omega[i]), \kappa[i], i)\right)\right). \tag{4}$$

Observe that this function closely resembles TRIENC (1). More concretely, for any $\omega$ and $i$, we have that $\zeta(\omega[i]) = \boldsymbol{h}_{\omega[i]}$, and, moreover, $\boldsymbol{p}_{\kappa[i]}$ and $\boldsymbol{p}_i$ are the positional encodings of $\kappa[i]$, and $i$, respectively. Hence, it is easy to see that there exists a function $\mu$, which satisfies, for every $i$:

$$((\zeta(\omega[i])), \kappa[i], i) = \mu(\boldsymbol{h}_{\omega[i]} \| \boldsymbol{p}_{\kappa[i]} \| \boldsymbol{p}_i).$$

This implies the following:

$$g(S_C) = \phi\left(\sum_{i=1}^{|\omega|} f\left((\zeta(\omega[i]), \kappa[i], i)\right)\right)$$
$$= \phi\left(\sum_{i=1}^{|\omega|} f\left(\mu(\boldsymbol{h}_{\omega[i]} \| \boldsymbol{p}_{\kappa[i]} \| \boldsymbol{p}_i)\right)\right)$$
$$= \phi\left(\sum_{i=1}^{|\omega|} (f \circ \mu)(\boldsymbol{h}_{\omega[i]} \| \boldsymbol{p}_{\kappa[i]} \| \boldsymbol{p}_i)\right).$$

Observe that this function can be parameterized by TRIENC: we apply the universal approximation theorem [14, 31, 32], and encode $(f \circ \mu)$ with an MLP (i.e., the inner MLP) and similarly encode $\phi$ with another MLP (i.e., the outer MLP).

This establishes a bijection $\rho$ between $\mathcal{F}$ and $\mathcal{M}$: for any SPQR component $C$, we can injectively map both the code $\text{CODE}(C)$ (or, equivalently the corresponding set $S_C$) and the representation $\widehat{\boldsymbol{h}}_C$ to the same unique value using the function $g$ as $\widehat{\boldsymbol{h}}_C = g(S_C)$, and we have shown that there exists a parameterization of TRIENC for this target function $g$.

**Parameterizing BIENC.** In this case, we are given a biconnected component $B$ and the representations $\widehat{h}_C$ of each component $C$ from the SPQR tree $\gamma = \text{SPQR}(B)$. We consider the representations $\widehat{h}_C$ which are a result of the parameterization of TRIENC, described earlier.

BIENC uses the SPQR tree and the integers $\theta(C, C')$ corresponding to how we connect $C$ and $C' \in \chi(C)$. BIENC then computes a representation for each subtree $\gamma_C$ induced by a node $C$ in a bottom up fashion as:

$$\widetilde{h}_{\gamma_C} = \text{MLP}\left(\widehat{h}_C + \sum_{C' \in \chi(C)} \text{MLP}\left(\widetilde{h}_{\gamma_{C'}} \| p_{\theta(C,C')}\right)\right). \tag{5}$$

This encoder operates in a bottom up fashion to ensure that a subtree representation of the children of $C$ exists before it encodes the subtree $\gamma_C$. The representation of the canonical root node in $\gamma$ is used as the representation of the biconnected component $B$ by setting: $\widetilde{h}_B = \widetilde{h}_{\gamma_{\text{ROOT}(\gamma)}}$.

To show the result, we first note that the canonical code given by the KHC algorithm also operates in a bottom up fashion on the subtrees of $\gamma$. We have two cases:

*Case 1.* For a *leaf node* $C$ in $\gamma$, the code for $\gamma_C$ is given by $\text{CODE}(\gamma_C) = \text{TRICODE}(C)$. This case can be seen as a special case of Case 2 (and we will treat it as such).

*Case 2.* For a *non-leaf node* $C$, we concatenate the codes of the subtrees induced by the children of $C$ in their lexicographical order, by first prepending the integer given by $\theta$ to each child code. Then, we also prepend the code of the SPQR component $C$ to this concatenation to get $\text{CODE}(\gamma_C)$. More precisely, if the lexicographical ordering of $\chi(u)$, based on $\text{CODE}(\gamma_{C'})$ for a given $C' \in \chi(C)$ is $x[1], \ldots, x[|\chi(C)|]$, then the code for $\gamma_C$ is given by:

$$\text{CODE}(\gamma_C) = \left(\text{TRICODE}(C), (\theta(C, x[1]), \text{CODE}(\gamma_{x[1]})), \ldots, (\theta(C, x[|x|]), \text{CODE}(\gamma_{x[|x|]}))\right) \tag{6}$$

We show a bijection between the multiset of codes $\mathcal{F}$ (given by the KHC algorithm), and the multiset of representations $\mathcal{M}$ (given by BIENC), where the respective multisets are defined, based on $G$ and the SPQR tree $\gamma$, as follows:

- $\mathcal{F} = \{\!\{\text{CODE}(\gamma_C) \mid C \in \gamma\}\!\}$: the multiset of all codes of all the induced SPQR subtrees.

- $\mathcal{M} = \{\!\{\widetilde{h}_{\gamma_C} \mid C \in \gamma\}\!\}$: the multiset of the representations of all the induced SPQR subtrees.

Analogously to the proof of TRIENC, we prove the following claim:

**Claim 2.** *There exists a bijection $\rho$ between $\mathcal{F}$ and $\mathcal{M}$ such that, for any node $C$ in $\gamma$:*

$$\rho(\widetilde{h}_{\gamma_C}) = \text{CODE}(\gamma_C) \text{ and } \rho^{-1}(\text{CODE}(\gamma_C)) = \widetilde{h}_{\gamma_C}$$

Given Claim 2, the result follows, since, using the bijection $\rho$, and the completeness of the codes generated by the KHC algorithm, we obtain:

$$\widetilde{h}_B \neq \widetilde{h}_{B'}$$
$$\Leftrightarrow$$
$$\widetilde{h}_{\gamma_{\text{ROOT}(B)}} \neq \widetilde{h}_{\gamma_{\text{ROOT}(B')}}$$
$$\Leftrightarrow$$
$$\rho^{-1}(\text{CODE}(\text{ROOT}(B))) \neq \rho^{-1}(\text{CODE}(\text{ROOT}(B')))$$
$$\Leftrightarrow$$
$$\text{CODE}(\text{ROOT}(B)) \neq \text{CODE}(\text{ROOT}(B'))$$
$$\Leftrightarrow$$
*$B$ and $B'$ are non-isomorphic.*

To prove Claim 2, let us first consider how $\text{CODE}(\gamma_C)$ is generated. For any node $C$ in $\gamma$, there is a bijection between the codes of the form given in Equation (6) and sets of the following form:

$$S_C = \{\!\{\text{TRICODE}(C)\}\!\} \cup \{\!\{(\theta(C, C'), \text{CODE}(\gamma_{C'})) \mid C' \in \chi(C)\}\!\} \tag{7}$$

Observe that the sets of this form are of bounded size (since the number of children is bounded) and each of their elements is from a countable set (given the size of the graph, we can also bound the number of different codes that can be generated). By Lemma A.1 there exists an injective map $g$ from such sets to the interval $[0, 1]$. Since the size of every such set is bounded, and every tuple is from a countable set, we can apply Lemma 5 of Xu et al. [63], and decompose the function $g$ as:

$$g(S_C) = \phi\left(\sum_{x \in S} f(x)\right),$$

for an appropriate choice of $\phi : \mathbb{R}^{d'} \to [0, 1]$ and $f(x) \in \mathbb{R}^{d'}$. Based on the structure of the elements of $S_C$, we can further rewrite this as follows:

$$g(S_C) = \phi\left(f(\text{TriCode}(C)) + \sum_{C' \in \chi(C)} f\left((\theta(C, C'), \text{Code}(\gamma_{C'}))\right)\right). \tag{8}$$

Observe the connection between this function and BiEnc (5): for every $C' \in \chi(C)$, we have $\widetilde{\boldsymbol{h}}_{\gamma_{C'}}$ instead of $\text{Code}(\gamma_C)$, and, moreover, $\boldsymbol{p}_{\theta(C,C')}$ is a positional encoding of $\theta(C, C')$. Then, there exists a function $\mu$ such that:

$$(\theta(C, C'), \text{Code}(\gamma_{C'})) = \mu(\boldsymbol{p}_{\theta(C,C')} \| \widetilde{\boldsymbol{h}}_{\gamma_{C'}}),$$

provided that the following condition is met:

$$\widetilde{\boldsymbol{h}}_{\gamma_{C'}} = \text{Code}(\gamma_{C'}) \text{ for any } C' \in \chi(C). \tag{9}$$

Importantly, the choice for $\mu$ can be the same for all nodes $C$. Hence, assuming the condition specified in Equation (9) is met, the function $g$ can be further decomposed using some function $f'(x) \in \mathbb{R}^{d'}$ which satisfies:

$$g(S_C) = \phi\left(f(\{\text{TriCode}(C)\}) + \sum_{C' \in \chi(C)} f\left((\theta(C, C'), \text{Code}(\gamma_{C'}))\right)\right)$$

$$= \phi\left(\widehat{\boldsymbol{h}}_C + \sum_{C' \in \chi(C)} f'\left(\mu\left(\boldsymbol{p}_{\theta(C,C')} \| \widetilde{\boldsymbol{h}}_{\gamma_{C'}}\right)\right)\right)$$

$$= \phi\left(\widehat{\boldsymbol{h}}_C + \sum_{C' \in \chi(C)} (f' \circ \mu)\left(\boldsymbol{p}_{\theta(C,C')} \| \widetilde{\boldsymbol{h}}_{\gamma_{C'}}\right)\right).$$

Observe that this function can be parameterized by BiEnc[6] (5): we apply the universal approximation theorem [14, 31, 32], and encode $(f' \circ \mu)$ with an MLP (i.e., the inner MLP) and similarly encode $\phi$ with another MLP (i.e., the outer MLP).

To conclude the proof of Claim 2 (and thereby the proof of Lemma 6.2), we need to show the existence of bijection $\rho$ between $\mathcal{F}$ and $\mathcal{M}$ such that, for any node $C$ in $\gamma$:

$$\rho(\widetilde{\boldsymbol{h}}_{\gamma_C}) = \text{Code}(\gamma_C) \text{ and } \rho^{-1}(\text{Code}(\gamma_C)) = \widetilde{\boldsymbol{h}}_{\gamma_C}.$$

This can be shown by a straight-forward induction on the structure of the tree $\gamma$. For the base case, it suffices to observe that $C$ is a leaf node, which implies $\widetilde{\boldsymbol{h}}_{\gamma_C} = \phi(\widehat{\boldsymbol{h}}_C)$ and $\text{Code}(\gamma_C) = \{\text{TriCode}(C)\}$. The existence of a bijection is then warranted by Claim 1. For the inductive case, assume that there is a bijection between the induced representations of the children of $C$ and their codes to ensure that the condition given in Equation (9) is met (which holds since the algorithm operates in a bottom up manner). Using the injectivity of $g$, and the fact that all subtree representations (of children) already admit a bijection, we can easily extend this to a bijection on all nodes $C$ of $\gamma$.

We have provided a parameterization of TriEnc and BiEnc and proven that they can compute representations which bijectively map to the codes of the KHC algorithm for the respective components, effectively aligning KHC algorithm with our encoders for these components. Given the completeness of the respective procedures in KHC, we conclude that the encoders are also complete in terms of distinguishing the respective components. $\qquad\square$

---

[6]Note that $f(\text{TriCode}(C))$ can be omitted, because TriEnc has an outer MLP, which can incorporate $f$.

Having showed that a single layer parameterization of BiEnc and TriEnc is sufficient for distinguishing the biconnected and triconnected components, we proceed with the main lemma.

**Lemma 6.3.** *For a planar graph $G = (V, E, \zeta)$ of order $n$ and its associated Block-Cut tree $\delta = \text{BlockCut}(G)$, there exists a $L = \lceil \log_2(n) \rceil + 1$ layer parameterization of BasePlane that computes a complete graph invariant for each subtree $\delta_u$ induced by each cut node $u$.*

*Proof.* CutEnc recursively computes the representation for induced subtrees $\delta_u$ from cut nodes $u$, where $\delta = \text{BlockCut}(G)$. Recall that in Block-Cut trees, the children of a cut node always represent a biconnected component, and the children of a biconnected component always represent a cut node. Therefore, it is natural to give the update formula for a cut node $u$ in terms of the biconnected component $B$ represented by $u$'s children $\chi(u)$ and $B$'s children $\chi(B)$.

$$\overline{\boldsymbol{h}}_{\delta_u}^{(\ell)} = \text{MLP}\left(\boldsymbol{h}_u^{(\ell-1)} + \sum_{B \in \chi(u)} \text{MLP}\left(\widetilde{\boldsymbol{h}}_B^{(\ell)} + \sum_{v \in \chi(B)} \overline{\boldsymbol{h}}_{\delta_v}^{(\ell)}\right)\right). \tag{10}$$

To show the result, we first note that the canonical code given by the KHC algorithm also operates in a bottom up fashion on the subtrees of $\delta$. We have three cases:

**Case 1.** For a *leaf* $B$ in $\delta$, the code for $\delta_B$ is given by $\text{CODE}(\delta_B) = \text{BiCODE}(B)$. This is because the leafs of $\delta$ are all biconnected components.

**Case 2.** For a *non-leaf* biconnected component $B$ in $\delta$, we perform overrides for the codes associated with each child cut node, and then use BiEnc. More precisely, we override the associated $\text{CODE}(\{\{u\}, \{\}\}) := \text{CODE}(\delta_u)$ for all $u \in \chi(B)$, and then we compute $\text{CODE}(\delta_B) = \text{BiCODE}(B)$.

**Case 3.** For a *non-leaf* cut node $u$ in $\delta$, we encode in a similar way to BiEnc: we get the set of codes induced by the children of $u$ in their lexicographical order. More precisely, if the lexicographical ordering of $\chi(u)$, based on $\text{CODE}(\delta_B)$ for a given $B \in \chi(u)$ is $x[1], \ldots, x[|\chi(u)|]$, then the code for $\delta_u$ is given by:

$$\text{CODE}(\delta_u) = \big(\text{CODE}(\delta_{x[1]})), \ldots, \text{CODE}(\delta_{x[|x|]})\big) \tag{11}$$

Instead of modelling the overrides (as in Case 2), BasePlane learns the cut node representations. We first prove this result by giving a parameterization of BasePlane which uses linearly many layers in $n$ and then show how this construction can be improved to use logarithmic number of layers. Specifically, we will first show that BasePlane can be parameterized to satisfy the following properties:

1. For every cut node $u$, we reserve a dimension in $\boldsymbol{h}_u^L$ that stores the number of cut nodes in $\delta_u$. This is done in the UPDATE part of the corresponding layer.

2. There is a bijection $\lambda$ between the representations of the induced subtrees and subtree codes, such that $\lambda(\overline{\boldsymbol{h}}_{\delta_u}^{(L)}) = \text{CODE}(\delta_u)$ *and* $\lambda^{-1}(\text{CODE}(\delta_u^{(L)})) = \overline{\boldsymbol{h}}_{\delta_u}^{(L)}$, for cut nodes $u$ that have strictly less than $L$ cut nodes in $\delta_u$.

3. There is a bijection $\rho$ between the cut node representations and subtree codes, such that $\rho(\boldsymbol{h}_u^{(L)}) = \text{CODE}(\delta_u)$ *and* $\rho^{-1}(\text{CODE}(\delta_u^{(L)})) = \boldsymbol{h}_u^{(L)}$, for cut nodes $u$ that have strictly less than $L$ cut nodes in $\delta_u$.

Observe that the property (2) directly gives us a complete graph invariant for each subtree $\delta_u$ induced by each cut node $u$, since the codes for every induced subtree are complete, and through the bijection, we obtain complete representations. The remaining properties are useful for later in order to obtain a more efficient construction.

The UPDATE function in every layer is crucial for our constructions, and we recall its definition:

$$\boldsymbol{h}_u^{(\ell)} = f^{(\ell)}\Big( g_1^{(\ell)}\big(\boldsymbol{h}_u^{(\ell-1)} + \sum_{v \in N_u} \boldsymbol{h}_v^{(\ell-1)}\big) \parallel g_2^{(\ell)}\big(\sum_{v \in V} \boldsymbol{h}_v^{(\ell-1)}\big) \parallel$$
$$g_3^{(\ell)}\big(\boldsymbol{h}_u^{(\ell-1)} + \sum_{C \in \sigma_u^G} \widehat{\boldsymbol{h}}_C^{(\ell)}\big) \parallel g_4^{(\ell)}\big(\boldsymbol{h}_u^{(\ell-1)} + \sum_{B \in \pi_u^G} \widetilde{\boldsymbol{h}}_B^{(\ell)}\big) \parallel \overline{\boldsymbol{h}}_{\delta_u}^{(\ell)}\Big),$$

We prove that there exists a parameterization of BASEPLANE which satisfies the properties (1)-(3) by induction on the number of layers $L$.

**Base case.** $L = 1$. In this case, there are no cut nodes satisfying the constraints, and the model trivially satisfies the properties (2)–(3). To satisfy (1), we can set the inner MLP in CUTENC as the identity function, and the outer MLP as a function which adds 1 to the first dimension of the input embedding. This ensures that the representations $\overline{\boldsymbol{h}}_{\delta_u}^{(1)}$ of cut nodes $u$ have their first components equal to the number of cut nodes in $\delta_u$. We can encode the property (1) in $\boldsymbol{h}_u^{(\ell)}$ using the representation $\overline{\boldsymbol{h}}_{\delta_u}^{(1)}$, since the latter is a readout component in UPDATE.

**Inductive step.** $L > 1$. By induction hypothesis, there is an $(L-1)$-layer parameterization of BASEPLANE which satisfies the properties (1)–(3). We can define the $L$-th layer so that our $L$ layer parameterization of BASEPLANE satisfies all the properties:

*Property (1):* For every cut node $u$, the function UPDATE has a readout from $\boldsymbol{h}_u^{(L-1)}$ and $\overline{\boldsymbol{h}}_{\delta_u}^{(L)}$, which allows us to copy the first dimension of $\boldsymbol{h}_u^{(L-1)}$ into the first dimension of $\boldsymbol{h}_u^{(L)}$ using a linear transformation, which immediately gives us the property.

*Property (2):* For this property, let us first consider the easier direction. Given the code of $\delta_u$, we want to find the CUTENC representation for the induced subtree of $u$. From the subtree code, we can reconstruct the induced subtree $\delta_u$, and then run a $L$-layer BASEPLANE on reconstructed Block-Cut Tree to find the CUTENC representation. As for the other direction, we want to find the subtree code of given the representation of the induced subgraph. The CUTENC encodes a multiset $\{\!\!\{(\widetilde{\boldsymbol{h}}_B, \{\!\!\{\boldsymbol{h}_v | v \in \chi(B)\}\!\!\}) \mid B \in \chi(u)\}\!\!\}$. By induction hypothesis, we know that all grandchildren $v \in \chi^2(u)$ already have properties (1)–(3) satisfied for them with the first $(L-1)$ layers. Hence, using the parameterization of TRIENC and BIENC given in Lemma 6.2, as part of the $L$-th BASEPLANE layer, we can get a bijection between BICODE and biconnected component representation. This way we can obtain BICODE$(B)$ for all the children biconnected components $B \in \chi(u)$, with all the necessary overriding. Having all necessarily overrides is crucial, because to get the KHC code for the cut node $u$, we need to concatenate the biconnected codes from 6 that already have the required overrides. Hence, we make the parameterization of CUTENC encode multisets of representations for biconnected components $B \in \chi(u)$, and by similar arguments as in the proof of Lemma 6.2, this can be done using the MLPs in CUTENC.

*Property (3):* Using the bijection from (2) as a bridge, we can easily show the property (3). In the update formula, we appended $\overline{\boldsymbol{h}}_{\delta_u}^{(L)}$ using the MLP. If the MLP is bijective with the dimension taken by $\overline{\boldsymbol{h}}_{\delta_u}^{(L)}$, we get a bijection between the node representation and the subtree representation. By transitivity, we get a bijection between node representations and subtree codes.

This concludes our construction using $L = O(n)$ BASEPLANE layers.

**Efficient construction.** We now show that the presented construction can be made more efficient, using only $\lceil \log_2(n) \rceil + 1$ layers. This is achieved by treating the cut nodes $u$ differently based on the number of cut nodes they include in their induced subtrees $\delta_u$. In this construction, the property (1) remains the same, but the properties (2)–(3) are modified:

1. For every cut node $u$, we reserve a component in $\boldsymbol{h}_u^{(L)}$ that stored to the number of cut nodes in $\delta_u$. This is done in the UPDATE part of the corresponding layer.

2. There is a bijection $\lambda$ between the representations of the induced subtrees and subtree codes, such that $\lambda(\overline{\boldsymbol{h}}_{\delta_u}^{(L)}) = \text{CODE}(\delta_u)$ *and* $\lambda^{-1}(\text{CODE}(\delta_u^{(L)})) = \overline{\boldsymbol{h}}_{\delta_u}^{(L)}$, for cut nodes $u$ that have strictly less than $2^{(L-1)}$ cut nodes in $\delta_u$.

3. There is a bijection $\rho$ between the cut node representations and subtree codes, such that $\rho(\boldsymbol{h}_u^{(L)}) = \text{CODE}(\delta_u)$ *and* $\rho^{-1}(\text{CODE}(\delta_u^{(L)})) = \boldsymbol{h}_u^{(L)}$, for cut nodes $u$ that have strictly less than $2^{(L-1)}$ cut nodes in $\delta_u$.

These new modified properties allow us to reduce the depth of the induction to a logarithmic number of steps, by having a more carefully designed parameterization of CUTENC. The logarithmic depth comes immediately from properties (2) and (3), while the core observations for building the CUTENC parameterizations are motivated by standard literature on efficient tree data structures, and in particular by the Heavy Light Decomposition (HLD) [55]. In HLD, we build "chains" through the tree that always go to the child that has the most nodes in its subtree. This gives us the property that, the path between each two nodes visits at most a logarithmic number of different chains, and we will use this concept in our parametrization. More precisely, we can construct a single CUTENC layer, that will make the above properties hold for whole chains, and not only individual nodes as in the previous construction.

We will once again use induction to show that such parametrizations exist.

**Base case.** $L = 1$. The properties are equivalent to the ones in the less efficient construction: property (1) is the same, and $2^{(L-1)} = 2^0 = L$, so we will satisfy them with the same parameterization.

**Inductive step.** $L > 1$. By induction hypothesis, there is an $(L-1)$-layer parameterization of BASEPLANE which satisfies the properties (1)–(3). We can define the $L$-th layer so that our $L$ layer parameterization of BASEPLANE satisfies all the properties:

*Property (1):* This is satisfied in the same way as in the less efficient construction - we allocate one of the dimensions for the required count, and propagate it from the previous layer.

*Property (2):* Let us consider an arbitrary cut node $u$ that has strictly less than $2^{(L-1)}$ cut nodes in its induced subtree $\delta_u$, and still does *not* satisfy property (2). We will call such nodes "improper" with respect to the current induction step, and show that there is a single layer CUTENC parameterization that makes all improper nodes satisfy these two properties, or become "proper".

First, observe that an arbitrary cut node $u = v_0$ has at most one improper grandchild $v_1 \in \chi^2(u)$, because if there were more improper grandchildren then it would have had at least $2^{(L-1)}$ cut nodes in it. Repeating a similar argument, we know that there is at most one improper $v_2 \in \chi^2(v_1)$. Continuing this until there are no improper grandchildren, we form a sequence of improper cut nodes $u = v_0, v_1, \ldots, v_k$, such that $v_{i+1}$ is a grandchild of $v_i$ for all $0 \le i < k$. The chain $u = v_0, v_1, \ldots, v_k$ is precisely a chain in HLD, and we will show that a single CUTENC layer can make an "improper chain" satisfy the properties. We prove this by applying inner induction on the chain:

**Induction hypothesis (for grandchildren):** By the induction hypothesis, we have an established $(L-1)$-layer parameterization of BASEPLANE that satisfies properties (1)–(3) for all but a single grandchild of *each* node in the chain $v_0, ..., v_k$. This assumption is critical as it provides the ground truth that our subsequent steps will build upon.

**Inner induction:** Having established the hypothesis for our node's children, we initiate an inner induction, starting from the farthest improper node in the chain, $v_k$, and progressively moving towards the root of the chain, $u = v_0$. This process is akin to traversing back up the chain, ensuring each node, starting from $v_k$, satisfies property (2) as we ascend.

For each node $v_i$ in our reverse traversal, we apply the special property: there exists at most one improper grandchild (potentially $v_{i+1}$) within its local structure. Our CUTENC architecture then identifies this improper grandchild, which, crucially, is "proper" in its representation due to our inner induction process. This *identification* is facilitated by the inner MLP given in Equation (10), and property (1), as identifying is equivalent to selecting the grandchild with the largest number of cut nodes in the allocated dimension.

Having identified and isolated our improper grandchild $v_{i+1}$ (now proper in $\bar{\boldsymbol{h}}_{\delta_{v_{i+1}}}^{(L)}$), CUTENC then integrates this node with the other children and grandchildren of $v_i$. It is important to note that all these other nodes have representations bijective to their respective subtree *codes* because of the induction hypothesis. This integration is not a mere collection; instead, the outer MLP in

Equation (10) models the "code operation" that BICODE would have done, which is possible due to the mentioned bijections to codes. This concludes the inner induction.

Our choice for $u$ was arbitrary and the required CUTENC parameterization is independent from the exact node representaton - it simply models a translation back to codes, and then injecting back to embeddings. Hence, all chains of improper nodes will satisfy property (2) after a single CUTENC layer.

*Property (3):* We once again follow the same approach as in the less efficient construction - we use property (2) as a bridge, as we know we have already satisfied it. Property (3) is needed to ensure that all nodes that once became "proper" will always stay so, by propagating the bijection to codes.

This concludes our construction using $\lceil \log_2(n) \rceil + 1$ layers. $\qquad \square$

**Theorem 6.1.** *For any planar graphs $G_1 = (V_1, E_1, \zeta_1)$ and $G_2 = (V_2, E_2, \zeta_2)$, there exists a parameterization of* BASEPLANE *with at most $L = \lceil \log_2(\max\{|V_1|, |V_2|\}) \rceil + 1$ layers, which computes a complete graph invariant, that is, the final graph-level embeddings satisfy $z_{G_1}^{(L)} \neq z_{G_2}^{(L)}$ if and only if $G_1$ and $G_2$ are not isomorphic.*

*Proof.* The "only if" direction is immediate because BASEPLANE is an invariant model for planar graphs. To prove the "if" direction, we do a case analysis on the root of the two Block-Cut Trees. For each case, we provide a parameterization of BASEPLANE such that $z_{G_1}^{(L)} \neq z_{G_2}^{(L)}$ for any two non-isomorphic graphs $G_1$ and $G_2$. A complete BASEPLANE model can be obtained by appropriately unifying the respective parameterizations.

**Case 1.** ROOT($\delta_1$) and ROOT($\delta_2$) represents two cut nodes.

Consider a parameterization of the final BASEPLANE update formula, where only cut node representation is used, and a simplified readout that only aggregates from the last layer. We can rewrite the readout for a graph in terms of the cut node representation from the last BASEPLANE layer:

$$z_G^{(L)} = \text{MLP}\left( \sum_{u \in V^G} \text{MLP}(\bar{h}_{\delta_u}^{(L)}) \right).$$

Let $\text{CUT}(G) = \{\!\{\bar{h}_{\delta_u}^{(L)} \mid u \in V^G\}\!\}$. Intuitively, $\text{CUT}(G)$ is a multiset of cut node representations from the last BASEPLANE layer. We assume $|V_{\delta_1}| \leq |V_{\delta_2}|$ without loss of generality. Consider the root node ROOT($\delta_2$) of the Block-Cut Tree $\delta_2$. By Lemma 6.3, we have $h_{\text{ROOT}(\delta_2)}$ as a complete graph invariant with $L$ layers. Since $\delta_2$ cannot appear as a subtree of $\delta_1$, $h_{\text{ROOT}(\delta_2)} \notin \text{CUT}(G_1)$. Hence, $\text{CUT}(G_1) \neq \text{CUT}(G_2)$. Since this model can define an injective mapping on the multiset $\text{CUT}(G)$ using similar arguments as before, we get that $z_{G_1}^{(L)} \neq z_{G_2}^{(L)}$.

**Case 2.** ROOT($\delta_1$) and ROOT($\delta_2$) represents two biconnected components.

We use a similar strategy to prove Case 2. First, we consider a simplified BASEPLANE model, where the update formula considers biconnected components only and the final readout aggregates from the last BASEPLANE layer. We similarly give the final graph readout in terms of the biconnected component representation from the last BASEPLANE layer.

$$z_G = \text{MLP}\left( \sum_{u \in V^G} \text{MLP}((h_u^{(L-1)} + \sum_{B \in \pi_u^G} \widetilde{h}_B^{(L)})) \right).$$

Let $\text{BC}(G) = \{(h_u^{(L-1)}, \{\!\{\widetilde{h}_B^{(L)} \mid B \in \pi_u^G\}\!\}) \mid u \in V^G\}$. In Lemma 6.3, we prove that $\widetilde{h}_B^{(L)}$ is also a complete invariant for the subtree rooted at $B$ in the Block-Cut Tree. First, we show $\text{BC}(G_1) \neq \text{BC}(G_2)$. As before, we assume $|V_{\delta_1}| \leq |V_{\delta_2}|$ without loss of generality. Consider how

the biconnected component representation $\boldsymbol{h}_{\text{ROOT}(\delta_2)}$ appears in the two multisets of pairs. For $G_2$, there exist at least one node $u$ and pair:

$$(\boldsymbol{h}_u^{(L-1)}, \{\!\!\{ \widetilde{\boldsymbol{h}}_B^{(L)} \mid B \in \pi_u^G \}\!\!\}) \in \text{BC}(G_2),$$

such that $\boldsymbol{h}_{\text{ROOT}(\delta_2)} \in \{\!\!\{ \widetilde{\boldsymbol{h}}_B^{(L)} \mid B \in \pi_u^G \}\!\!\}$. However, because $\boldsymbol{h}_{\text{ROOT}(\delta_2)}$ is a complete invariant for $\delta_2$ and $\delta_2$ cannot appear as a subtree in $\delta_1$, no such pair exists in $\text{BC}(G_1)$. Given $\text{BC}(G_1) \neq \text{BC}(G_2)$, we can parameterize the MLPs to define an injective mapping to get that $\boldsymbol{z}_{G_1}^{(L)} \neq \boldsymbol{z}_{G_2}^{(L)}$.

**Case 3.** $\text{ROOT}(\delta_1)$ represents a cut node and $\text{ROOT}(\delta_2)$ represents a biconnected component.

We can distinguish $G_1$ and $G_2$ using a simple property of $\text{ROOT}(\delta_1)$. Recall that $\pi_u^G$ represents the set of biconnected components that contains $u$, and $\chi^\delta(C)$ represents the C's children in the Block-Cut Tree $\delta$. For $\text{ROOT}(\delta_1)$, we have $|\pi_u^{G_1}| = |\chi^{\delta_1}(u)|$. However, for any other cut node $u$, including the non-root cut node in $\delta_1$ and all cut nodes in $\delta_2$, we have $|\pi_u^G| = |\chi^\delta(u)| + 1$, because there must be a parent node $v$ of $u$ such that $v \in \pi_u^{G_1}$ but $v \notin \chi^\delta(u)$.

Therefore, we consider a parameterization of a one-layer BASEPLANE model that exploits this property. In BIENC, we have a constant vector $[1, 0]^\top$ for all biconnected components. In CUTENC, we learn $[0, |\chi_u|]^\top$ for all cut nodes $u$. In the update formula, we have $[|\pi_u^G| - |\chi^\delta(u)| - 1, 0]^\top$. All of the above specifications can be achieved using linear maps. For the final readout, we simply sum all node representations with no extra transformation.

Then, for $\text{ROOT}(\delta_1)$, we have $\boldsymbol{h}_{\text{ROOT}(\delta_u)} = [-1, 0]^\top$. For any other cut node $u$, we have $\boldsymbol{h}_u = [0, 0]^\top$. For all non-cut nodes $u$, we also have $\boldsymbol{h}_u = [0, 0]^\top$ because $|\pi_u^G| = 1$ and $\bar{\boldsymbol{h}}_{\delta_u} = [0, 0]^\top$. Summing all the node representations yields $\boldsymbol{z}_{G_1} = [-1, 0]^\top$ but $\boldsymbol{z}_{G_2} = [0, 0]^\top$. Hence, we obtain $\boldsymbol{z}_{G_1}^{(L)} \neq \boldsymbol{z}_{G_2}^{(L)}$, as required. $\qquad\square$

## B  Runtime analysis of BASEPLANE

### B.1  Asymptotic analysis

In this section, we study the runtime complexity of the BASEPLANE model.

**Computing components of a planar graph.** Given an input graph $G = (V, E, \zeta)$, BASEPLANE first computes the SPQR components/SPQR tree, and identifies cut nodes. For this step, we follow a simple $O(|V|^2)$ procedure, analogously to the computation in the KHC algorithm. Note that this computation only needs to run once, as a pre-processing step. Therefore, this pre-computation does not ultimately affect runtime for model predictions.

**Size and number of computed components.**

1. *Cut nodes:* The number of cut nodes in $G$ is at most $|V|$, and this corresponds to the worst-case when $G$ is a tree.

2. *Biconnected Components:* The number of biconnected components $|\pi^G|$ is at most $|V| - 1$, also obtained when $G$ is a tree. This setting also yields a worst-case bound of $2|V| - 2$ on the total number of nodes across all individual biconnected components.

3. *SQPR components:* As proved by Gutwenger and Mutzel [27], given a biconnected component $B$, the number of corresponding SPQR components, as well as their size, is bounded by the number of nodes in $B$. The input graph may have multiple biconnected components, whose total size is bounded by $2|V| - 2$ as described earlier. Thus, we can apply the earlier result from Gutwenger and Mutzel [27] to obtain an analogous bound of $2|V| - 2$ on the total number of SPQR components.

4. *SPQR trees*: As each SPQR component corresponds to exactly one node in a SPQR tree. The total size of all SPQR trees is upper-bounded by $2|V| - 2$.

**Complexity of TRIENC.** TRIENC computes a representation for each SPQR component edge. This number of edges, which we denote by $e_C$, is in fact linear in $V$: Indeed, following Euler's theorem for planar graphs ($|E| \leq 3|V| - 6$), the number of edges per SPQR component is linear in its size.

Moreover, since the total number of nodes across all SPQR components is upper-bounded by $2|V| - 2$, the total number of edges is in $O(V)$. Therefore, as each edge representation can be computed in a constant time with an MLP, TRIENC computes all edge representations across all SPQR components in $O(e_C \cdot d) = O(|V|d^2)$ time, where $d$ denotes the embedding dimension of the model.

Using the earlier edge representations, TRIENC performs an aggregation into a triconnected component representation by summing the relevant edge representations, and this is done in $O(e_C d)$. Then, the sum outputs across all SPQR components are transformed using an MLP, in $O(|\sigma^G|d^2)$. Therefore, the overall complexity of TRIENC is $O((e_c + |\sigma^G|)d^2) = O(|V|d^2)$.

**Complexity of BIENC.** BIENC recursively encodes nodes in the SPQR tree. Each SPQR node is aggregated once by its parent and an MLP is applied to each node representation once. The total complexity of this call is therefore $O(|\sigma^G|d^2) = O(|V|d^2)$.

**Complexity of CUTENC.** A similar complexity of $O(|V|d^2)$ applies for CUTENC, as CUTENC follows a similar computational pipeline.

**Complexity of node update.** As in standard MLPs, message aggregation runs in $O(|E|d)$, and the combine function runs in $O(|V|d^2)$. Global readouts can be computed in $O(|V|d)$. Aggregating from bi-connected components also involves at most $O(|V|d)$ messages, as the number of messages corresponds to the total number of bi-connected component nodes, which itself does not exceed $2|V| - 2$. The same argument applies to messages from SPQR components: nodes within these components will message their original graph analogs, leading to the same bound. Finally, the cut node messages are linear, i.e., $O(|V|)$, as each node receives exactly one message (no aggregation, no transformation). Overall, this leads to the step computation having a complexity of $O(|V|d^2)$.

**Overall complexity.** Each BASEPLANE layer runs in $O(|V|d^2)$ time, as this is the asymptotic bound of each of its individual steps. The $d^2$ term primarily stems from MLP computations, and is not a main hurdle to scalability, as the used embedding dimensionality in our experiments is usually small.

**Parallelization.** Parallelization can be used to speed up the pre-processing of large planar graphs. In particular, parallelization can reduce the runtime of KHC pre-processing, and more precisely the computation of the canonical walk in Weinberg's algorithm, which itself is the individual step with the highest complexity ($O(|V|^2)$) in our approach. To this end, one can independently run Weinberg's algorithm across several triconnected components in parallel. Moreover, we can also parallelize the algorithm's operation within a single triconnected component: as the quadratic overhead comes from computing the lexicographically smallest TRICODE, where the computation depends on the choice of the first walk edge, we can concurrently evaluate each first edge option and subsequently aggregate over all outputs to more efficiently find the smallest code.

## B.2 Empirical runtime evaluation

**Experimental setup.** To validate the runtime efficiency of BASEPLANE, we conduct a runtime experiment on real-world planar graphs based on the geographic faces of Alaska from the dataset TIGER, provided by the U.S. Census Bureau. The original dataset is TIGER-Alaska-93K and has 93366 nodes. We also extract the smaller datasets TIGER-Alaska-2K and TIGER-Alaska-10K, which are subsets of the original dataset with 2000 and 10000 nodes, respectively. We compare the wallclock time of BASEPLANE and PPGN which is as expressive as 2-WL.

Table 7: Runtime experiments on planar graphs TIGER-Alaska-2K, TIGER-Alaska-10K, and TIGER-Alaska-93K. We report the pre-processing and training time (per epoch) for all models.

| | | BASEPLANE | | PPGN | | ESAN | |
|---|---|---|---|---|---|---|---|
| Dataset | #Nodes | Pre. | Train | Pre. | Train | Pre. | Train |
| TIGER-Alaska-2K | 2000 | 9.8 sec | 0.1 sec | 3.4 sec | 5.9 sec | 4.6 sec | 87.73 sec |
| TIGER-Alaska-10K | 10000 | 50 sec | 0.33 sec | OOM | OOM | OOM | OOM |
| TIGER-Alaska-93K | 93366 | 3.7 h's | 2.2 sec | OOM | OOM | OOM | OOM |

**Results.** The runtimes reported in Table 7 confirm our expectation: BASEPLANE is the only model which can scale up to all datasets. Indeed, both ESAN and PPGN fail on TIGER-Alaska-10K and TIGER-Alaska-93K. Somewhat surprisingly, ESAN training time is slower than PPGN training time

Table 8: BASEPLANE ablation study on ZINC. Each component is essential to achieve the reported performance. The use of bi- and triconnected components yields substantial improvements. Aggregating over direct neighbours in the update equation remains important.

| Models | ZINC MAE |
|---|---|
| BASEPLANE (original) | **0.076**± 0.003 |
| BASEPLANE (no readout) | 0.079± 0.002 |
| BASEPLANE (no CUTENC) | 0.079± 0.003 |
| BASEPLANE (no neighbours) | 0.099± 0.004 |
| BASEPLANE (only BIENC) | 0.097± 0.002 |
| BASEPLANE (only TRIENC) | 0.092± 0.003 |

on TIGER-Alaska-2K which is likely due to the fact that the maximal degree of TIGER-Alaska-2K is 620, which the ESAN algorithm depends on. BASEPLANE is very fast for training with the main bottleneck being *one-off* pre-processing. This highlights that BASEPLANE is a highly efficient option for inference on large-scale graphs compared to subgraph or higher-order alternatives.

## C  An ablation study on ZINC

We additionally conduct extensive ablation studies on each component in our model update equation using the ZINC 12k dataset. We report the final results in Table 8. The setup and the main findings can be summarized as follows:

- BASEPLANE (no readout): We removed the global readout term from the update formula and MAE worsened by 0.003.

- BASEPLANE (no CutEnc): We removed the cut node term from the update formula and MAE worsened by 0.003.

- BASEPLANE (no neighbors): We additionally removed the neighbor aggregation from the update formula and MAE worsened by 0.023.

- BASEPLANE (only BiEnc): We only used the triconnected components for the update formula and MAE worsened by 0.021.

- BASEPLANE (only TriEnc): We only used the triconnected components for the update formula and MAE worsened by 0.016.

Hence, BasePlanE significantly benefits from each of its components: the combination of standard message passing with component decompositions is essential to obtaining our reported results.

## D  Further experimental details

### D.1  Link to code

The code for our experiments, as well as instructions to reproduce our results and set up dependencies, can be found at this GitHub repository: `https://github.com/ZZYSonny/PlanE`

### D.2  Computational resources

We run all experiments on 4 cores from Intel Xeon Platinum 8268 CPU @ 2.90GHz with 32GB RAM. In Table 9, we report the approximate time to train a BASEPLANE model on each dataset and with each tuned hidden dimension value.

### D.3  E-BASEPLANE architecture

E-BASEPLANE builds on BASEPLANE, and additionally processes edge features within the input graph. To this end, it supersedes the original BASEPLANE update equations for SPQR components

Table 9: Approximate Training Time for BASEPLANE.

| Dataset Name | Hidden Dimension | Training Time (hours) |
|---|---|---|
| QM9$_{\text{CC}}$ | 32 | 2.5 |
| MolHIV | 64 | 6 |
| QM9 | 128 | 25 |
| ZINC(12k) | 64 | 5 |
|  | 128 | 8 |
| ZINC(Full) | 128 | 45 |
| EXP | 32 | 0.2 |

and nodes with the following analogs:

$$\widehat{\boldsymbol{h}}_C^{(\ell)} = \text{MLP}\Big(\sum_{i=1}^{|\omega|} \text{MLP}(\boldsymbol{h}_{\omega[i]}^{(\ell-1)}\|\hat{\boldsymbol{h}}_{\omega[i],\omega[(i+1)\%|w|]}^{(\ell-1)}\|\boldsymbol{p}_{\kappa[i]}\|\boldsymbol{p}_i))\Big), \text{ and}$$

$$\boldsymbol{h}_u^{(\ell)} = f^{(\ell)}\Big(g_1^{(\ell)}\big(\boldsymbol{h}_u^{(\ell-1)} + \sum_{v\in N_u} g_5^{(\ell)}(\boldsymbol{h}_v^{(\ell-1)}\|\boldsymbol{h}_{v,u}^{(\ell-1)})\big) \| g_2^{(\ell)}\big(\sum_{v\in V} \boldsymbol{h}_v^{(\ell-1)}\big)$$

$$g_3^{(\ell)}\big(\boldsymbol{h}_u^{(\ell-1)} + \sum_{B\in\pi_u^G} \widetilde{\boldsymbol{h}}_B^{(\ell)}\big) \| g_4^{(\ell)}\big(\boldsymbol{h}_u^{(\ell-1)} + \sum_{C\in\sigma_u^G} \widehat{\boldsymbol{h}}_C^{(\ell)}\big) \| \bar{\boldsymbol{h}}_{\delta_u}^{(\ell)}\Big),$$

where $\hat{\boldsymbol{h}}_{i,j} = \boldsymbol{h}_{i,j}$ if $(i,j) \in E$ and is the ones vector $(\boldsymbol{1}^d)$ otherwise

### D.4 Training setups

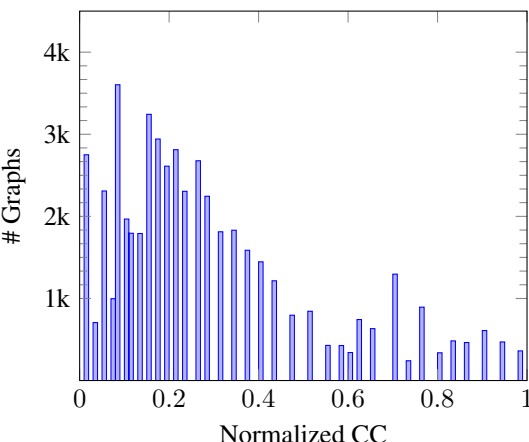

Figure 5: Normalized clustering coefficient distribution of QM9$_{\text{CC}}$.

**Graph classification on EXP** We follow the same protocol as the original work: we use 10-fold cross validation on the dataset, train BASEPLANE on each fold for 50 epochs using the Adam [39] optimizer with a learning rate of $10^{-3}$, and binary cross entropy loss.

**Graph classification on P3R.** We follow a very similar protocol to the one in EXP: we use 10-fold cross validation, where we train BASEPLANE for 100 epochs using the Adam [39] optimizer with a learning rate of $10^{-3}$, and cross entropy loss.

**Clustering coefficient of QM9 graphs.** To train all baselines, we use the Adam optimizer with a learning rate from $\{10^{-3}; 10^{-4}\}$, and train all models for 100 epochs using a batch size of 256 and L2 loss. We report the overall label distribution of normalized clustering coefficients on QM9 graphs in Figure 5.

**Graph classification on MolHIV.** We instantiate E-BASEPLANE with an embedding dimension of 64 and a positional encoding dimensionality of 16. We further tune the number of layers within the

set $\{2, 3\}$ and use a dropout probability from the set $\{0, 0.25, 0.5, 0.66\}$. Furthermore, we train our models with the Adam optimizer [39], with a constant learning rate of $10^{-3}$. Finally, we perform training with a batch size of 256 and train for 300 epochs.

**Graph regression on QM9.** As standard, we train E-BASEPLANE using mean squared error (MSE) and report mean absolute error (MAE) on the test set. For training E-BASEPLANE, we tune the learning rate from the set $\{10^{-3}, 5 \times 10^{-4}\}$ with the Adam optimizer, and adopt a learning rate decay of 0.7 every 25 epochs. Furthermore, we use a batch size of 256, 128-dimensional node embeddings, and 32-dimensional positional encoding.

**Graph regression on ZINC.** In all experiments, we use a node embedding dimensionality from the set $\{64, 128\}$, use 3 message passing layers, and 16-dimensional positional encoding vectors. We train both BASEPLANE and E-BASEPLANE with the Adam optimizer [39] using a learning rate from the set $\{10^{-3}, 5 \times 10^{-4}, 10^{-4}\}$, and follow a decay strategy where the learning rate decays by a factor of 2 for every 25 epochs where validation loss does not improve. We train using a batch size of 256 in all experiments, and run training using L1 loss for 500 epochs on the ZINC subset, and for 200 epochs on the full ZINC dataset.

