# OpenReview forum: "PlanE: Representation Learning over Planar Graphs"
_NeurIPS.cc/2023/Conference — NeurIPS 2023 poster_

### Official Review · Reviewer_Wih2 · 2023-06-21

**Soundness:** 4 excellent
**Presentation:** 3 good
**Contribution:** 3 good
**Rating:** 6
**Confidence:** 4

**Summary:**

The paper proposes PLANE: a graph neural network which is complete on planar graphs. The idea behind this is very natural and well explained in the paper: while graphs in general are difficult to separate, and as a result standard GNNs cannot separate them, planar graphs can be separated in polynomial time, and are very common in applications. Therefore it is useful to have a complete algorithm for these graphs.

**Strengths:**

As noted above, the question addressed by the paper is very natural, and the provided method achieves good empirical results

**Weaknesses:**

I felt the algorithm was
(a) very technically involved and difficult to understand
and
(b) did not seem very novel, or very much like an architecture for graphs: that is, most neural networks are composed of simple building blocks (convolutions). This is not the case here. Rather it seems like the authors took a planar graph isomorphism algorithm, and then somewhat superficially replaced hashing operations in the algorithm with continuous set valued operations.

**Questions:**

Main question: The part about why we want GNNs complete for planar graphs is explained very well. What is not explained so well, I believe, is: given that there exist `graph isomorphism algorithms' for planar graphs, what is it that needs to be done to turn them into architectures, and what, in the high level, is your contribution here? What were the challenges?

Technical question: For a given graph, is the partition into maximal biconnected (or triconnected) components unique? What is the source for the information presented in lines 75-84 in page 2? No reference is given

Experimental question: could you report how well both E-BasePlane and BasePlane did on all tasks?

Minor typos and comments for authors use (no need to discuss this in rebuttal):

The usage of \mathbb{C} is a notation for anything other than complex numbers is confusing. I would suggest redefining this to something else (e.g,. \mathcal{C})

Page 4 line 171: what are `two node dipole graphs'? explain?

Page 7 line 278 `clustering coefficient'-what is this?

In Table 4 and Line 377 you use (E)-BasePlane and in other places E-BasePlane. If the adding of parenthesis is intentional you should explain what is meant by it.

In Table 4 bottom line there is a slanted font, different from the rest of the table. This also happens in the other tables occasionally. If this is intentional explain what it means, otherwise fix font to be consistent




**Limitations:**

Overall yes. It does not seem the method can handle node features, this could be added to the limitations.

---

> ### Author Rebuttal · Authors · 2023-08-09
>
> Thank you for your constructive feedback! We respond to your comments below:
>
> > Main question: The part about why we want GNNs complete for planar graphs is explained very well. What is not explained so well, I believe, is: given that there exist `graph isomorphism algorithms' for planar graphs, what is it that needs to be done to turn them into architectures, and what, in the high level, is your contribution here? What were the challenges?
>
> The fundamental challenge is to turn a classical algorithm which is tailored for isomorphism testing on planar graphs into a general learning algorithm on planar graphs. In our context, the goal is to capture similarities between the graph structures along with the features. In fact, a learning algorithm needs to implicitly compute structural similarities, which is a harder problem than plain isomorphism testing. More concretely, the challenges we encountered can be summarized as follows:
>
> (1) How to encode cut nodes, biconnected components, and triconnected components in order to adequately update these representations to best capture the structural similarities?
>
> KHC does not keep track of node-level or component level representations. The codes can be of varying lengths and do not capture a notion of similarity. In contrast to this, we need these representations to be explicitly learned, as they indicate node-wise, component-wise, or graph-wise similarities.
>
> (2) How to integrate classical operations into meaningful computational layers without leading to optimization problems?
>
> KHC employs standard operations such as overriding, or interleaving, which have no natural counterpart in a learning algorithm: this is due to the fact that codes do not need to reflect similarity, unlike the hidden representations. This constraints the design space of our algorithm: we had to replace the interleave operation (an operation on the Block cut tree nodes) with a multi-layer update, and show that this suffices for completeness: we have shown logarithmically many layers suffice.
>
> (3) What kind of inductive biases are helpful in a learning algorithm which are not relevant for code generation of KHC?
>
> There are operations in KHC which do not have a counterpart in PlanE, but the converse is also true: we consider node neighbor representations in the update for better local context and also a standard readout for better global context. The lack of these worsens our empirical results as it can be seen from our ablation study (Table 1 in response.pdf), whereas these have no importance for the KHC algorithm.
>
> All in all, we revisited the KHC planarity algorithm, modified it in several non-trivial ways to arrive at a provably complete, learnable and scalable model PlanE, and the main challenge was  designing components to preserve the theoretical properties of KHC while also having the right inductive biases for graph ML.
>
> > Technical question: For a given graph, is the partition into maximal biconnected (or triconnected) components unique? What is the source for the information presented in lines 75-84 in page 2? No reference is given
>
> Yes indeed, an undirected graph (even if not planar) can be decomposed into its maximal biconnected components in linear time and this decomposition is unique. This is from classical literature, e.g., *“Hopcroft and Tarjan (1973) Algorithm 447: efficient algorithms for graph manipulation.”* The same is true for triconnected components and SPQR trees, see, e.g., *“Gutwenger, C., Mutzel, P. (2001). A Linear Time Implementation of SPQR-Trees”* for a classical reference. We refer to these references and related literature regarding the definitions in the  mentioned paragraph.
>
> > Experimental question: could you report how well both E-BasePlane and BasePlane did on all tasks?
>
> As per your request, we also reran E-BasePlane without edge features (which is equivalent to BasePlanE) on QM9. Since it doesn't use edge features, the results worsened, but interestingly, the results remain nevertheless very strong compared to the baselines. For reference, for all target properties on QM9,  BasePlanE is on average only 5.6\% worse than E-BasePlanE. This shows a strong evidence on how well BasePlanE exploits structural information. We will report the results in the next version of our paper (as the table is very large to fit into our one-page rebuttal document)

---

> > ### Comment · Reviewer_Wih2 · 2023-08-13
> >
> > I'm happy with the reivewers rebuttal and retain my score of 6.

---

> > > ### Author Response · Authors · 2023-08-14
> > >
> > > Thanks for going through the rebuttal, and for your continued positive view of our paper. We will integrate the comparison between E-BasePlanE and BasePlanE on QM9 into the appendix of the new version of the paper, because we agree this is insightful for the reader. Please let us know if there are any remaining concerns.

---

### Official Review · Reviewer_o9ac · 2023-07-04

**Soundness:** 2 fair
**Presentation:** 2 fair
**Contribution:** 1 poor
**Rating:** 4
**Confidence:** 4

**Summary:**

The paper deals with supervised learning on the graphs. Conventional models of message passing Graph Neural Networks are known to be restricted in their expressivity by the 1-WL test for isomorphism. Although more expressive models like higher-order GNNs have been proposed, they are inefficient and also, are not able to generate complete invariants for all graphs. The authors in this paper, propose to specialise their study to a class of graphs for which complete invariants can be efficiently computed i.e. planar graphs. They propose “PLANE” framework based on the graph isomorphism algorithm for planar graphs. PLANE can learn complete invariants for planar graphs in an efficient way and experiments are shown to validate the promising nature of the proposed method.

**Strengths:**

1. The paper is well-motivated in finding complete invariants for a class of graphs i.e. planar graphs while remaining computationally efficient.
1. The paper is well-written with most parts of the paper easy to follow. However, few more illustrations would help the reader.
1. Experimental results are mixed, partly promising.

**Weaknesses:**

1. **Novelty:** The major problem with this paper is the lack of novel ideas or insights. It is well-known that planar graphs are solvable for isomorphism and algorithms exist to do that. This paper leverages the KHC algorithm in a straightforward manner. Presumably, simply running the KHC algorithm on graphs and using the generated codes as features would give similar results. What is the additional usefulness of learning almost exact same procedure with parameters is neither discussed nor shown with some empirical results.
1. There are no insights in the theoretical analysis as well. Improved expressivity and the logarithmic steps are directly adaptable from the planar isomorphism algorithm.
1. The experimental results are not sufficient to make a convincing case for the method.
1.    It is not very insightful to compare with GCN/GIN since they are known to be restricted with 1-WL power for clustering co-efficients or other experiments.
1.   However, if you could include other more expressive models and show improvement compared to these, that would be helpful. For example, 3-WL GNN(Maron et al. (2019)), PF-FGNN (Dupty et al. (2021)) should produce complete invariants as well. However, they are not specifically designed for planar graphs and PLANE method can give substantial improvements over these models on planar graphs, then it would make sense to use algorithms specialized for planar graphs.
1.   Other synthetic datasets like strongly regular planar graphs can be used to validate the method as well.
1.   Results on QM9 could be better analysed if comparison with other methods are provided or the units are same as used in Dimene​​t (Gasteiger et. al. (2019))

**References:**
+ [1] Maron et al.(2019). "Provably powerful graph networks." Advances in neural information processing systems 32
+ [2] Dupty et al. (2021) "PF-GNN: Differentiable particle filtering based approximation of universal graph representations." International Conference on Learning Representations.
+ [3] Gasteiger et al. (2019). Directional Message Passing for Molecular Graphs. In International Conference on Learning Representations.

**Overall**,

I find it hard to see significant contributions from this paper although the motivation of finding specialized learning models for planar graphs is interesting.

**Questions:**

Please address the weaknesses

**Limitations:**

Not applicable since this is general method on graphs.

---

> ### Author Rebuttal · Authors · 2023-08-09
>
> Thank you for your review. We address your concerns below.
>
> > Novelty: The major problem with this paper is the lack of novel ideas or insights….This paper leverages the KHC algorithm in a straightforward manner.”
>
> Our work generalizes the classical, KHC algorithm to a *learnable neural* model while maintaining completeness and theoretical guarantees. In effect, PlanE builds on KHC like MPNNs build on 1-WL. Indeed, MPNNs align with 1-WL while crucially learning features from data. PlanE achieves the same relative to KHC, with added complexity coming from mapping this more involved algorithm to learnable functions. This mapping is by no means straightforward, and requires a series of subtle decisions: Please see our response on our adaptation of KHC.
>
> > Presumably, simply running the KHC algorithm on graphs and using the generated codes as features would give similar results. What is the additional usefulness of learning almost exact same procedure with parameters is neither discussed nor shown with some empirical results.
>
> The learnability of PlanE is *critical* vs KHC, just as MPNN learnability is critical vs 1-WL. MPNNs vastly outperform 1-WL on almost all real-world datasets as they can learn dataset-specific features while still having the 1-WL inductive bias. We are *not* learning the KHC procedure with parameters, but rather learn *component representations* and a *specific feature mapping*, while also aligning with KHC.
>
> To make this more concrete, we conduct a dedicated ablation of BasePlanE on ZINC (cf. 	Table 1 of *response.pdf* and global response for details). There, we drop standard MPNN neighbor aggregation (which is not part of KHC), and show that BasePlanE degrades very significantly, despite being more faithful to KHC. We kindly ask the reviewer to take these into account.
>
> > There are no insights in the theoretical analysis as well. Improved expressivity and the logarithmic steps are directly adaptable from the planar isomorphism algorithm.
>
> Though PlanE builds on KHC, it does *not* emulate all its components one-to-one. For instance, we do not use the KHC interleaving component, as this does not map to a learnable function due to its intricate overrides (which overwrite component representations and *eliminate gradients*). Moreover, PlanE uses component and node representations, which are not in KHC, but are essential for learning due to local inductive bias.
>
> Given these fundamental differences, we had to provide *novel proofs* for all our results. As a case in point, KHC only requires *one* step due to interleaves, whereas we show a logarithmic step bound for PlanE based on our learnable components. We are happy to elaborate more on this.
>
> > The experimental results are not sufficient to make a convincing case for the method.
>
> To our knowledge, PlanE is the only method that is practically scalable and provably complete on planar graphs. Moreover, our results, obtained without any specific optimizations, are already competitive with specialized models, e.g., CIN on ZINC, and other methods, e.g., SPN on QM9.
>
> > It is not very insightful to compare with GCN/GIN since they are known to be restricted with 1-WL power for clustering co-efficients or other experiments.
>
> We include GCN/GIN as baselines, analogously to their use in standard 1-WL expressiveness tests. For stronger comparisons, we include an expressive subgraph baseline, ESAN, which we outperform by roughly 40%.
>
> > However, if you could include other more expressive models and show improvement compared to these, that would be helpful. For example, 3-WL GNN(Maron et al. (2019)), PF-FGNN (Dupty et al. (2021)) should produce complete invariants as well. However, they are not specifically designed for planar graphs and PLANE method can give substantial improvements over these models on planar graphs, then it would make sense to use algorithms specialized for planar graphs.
>
> Unfortunately, this statement seems the result of a common confusion between the dimension counts of FWL and WL: Folklore $k$-WL (or, $k$-FWL in ML literature) is equivalent to oblivious $(k+1)$-WL. Kiefer et al. (2019) show that 3-FWL (e.g., oblivious 4-WL) is complete on planar graphs, and this dimension remains the best known to date. Hence, higher-order models with an established *3-FWL* result are complete on planar graphs. However, both 3-WL-GNNs and PPGNs only have *2-FWL* expressive power. Moreover, PF-GNN provides no such expressiveness result. None of these models is provably complete on planar graphs. In fact, we are not aware of an implementation of a 3-FWL model: we are happy to incorporate such models provided there is one.
>
> > Other synthetic datasets like strongly regular planar graphs can be used to validate the method as well.
>
> Please note that we sought to use strongly regular graphs as in CWN, but there are only 7 strongly regular planar graphs, which are in fact $\text{K}_1$, $\text{K}_2$, $\text{K}_3$, $\text{K}_4$, $\text{C}_4$, $\text{C}_5$, and $\overline{3\text{K}_2}$. However, these graphs are easily distinguishable by standard GNNs, as all but $\text{C}_4$ and $\text{K}_4$ have a different number of nodes, and the latter pair have a different number of edges, rendering the whole task trivial.
>
> Instead, we propose a new synthetic dataset based on 3-regular planar graphs, and experiment with BasePlanE, GIN and PPGNs (see global response and Table 2 of *response.pdf*). There, GIN matches a random guess, while BasePlane and PPGNs perfectly solve the task. PPGNs can solve this task because these graphs are 2-FWL-distinguishable.
>
> > Results on QM9 could be better analysed if comparison with other methods are provided or the units are same as used in Dimene​​t (Gasteiger et. al. (2019))
>
> We follow the QM9 setup of Alon et al. In this version, the regression quantities are scaled for more uniformity, and the baselines used are trained using different grids/splits than Dimenet. Hence, these two setups are not comparable.

---

> > ### Comment · Reviewer_o9ac · 2023-08-16
> > **Reviewer response**
> >
> > Thank you for your response.
> >
> > 1. Novelty remains the main weakness of the paper. The justification that MPNN is analogous to 1-WL like the proposed algorithm with KHC is not reasonable. When early MPNN models were proposed like GCN/GIN, it provided insights which were unknown during the time. Saying that neural extensions are needed for similar algorithms is not convincing by itself. I’m not able find new insights so far.
> > From the KHC algorithm, it is clear that bi-connected and tri-connected components are the key to distinguishing planar graphs which was also shown in new experiments shared by authors. However, this is already known albeit independently of planar isomorphism [1].
> > I do not see additional novel insights we can get from the proposed approach.
> >
> > 1. Empirically, the model performs comparably to other SOTA methods, however is not better than them. There is benefit in the runtime being linear. However, as shown in [1], similar results can be achieved just by using biconnected components. Therefore, there is not much contribution in this regard.
> >
> > Overall, I appreciate the authors for providing additional experiments including time complexity. My concern on novelty and contributions remain. However, based on linear complexity with learning guarantees on planar graphs, I’m rising my score by one point.
> >
> > **References**
> >
> > + [1] Zhang, Bohang, et al. "Rethinking the Expressive Power of GNNs via Graph Biconnectivity." The Eleventh International Conference on Learning Representations. 2022.

---

> > > ### Author Response · Authors · 2023-08-17
> > >
> > > > Overall, I appreciate the authors for providing additional experiments including time complexity. My concern on novelty and contributions remain. However, based on linear complexity with learning guarantees on planar graphs, I’m rising my score by one point.
> > >
> > > Thank you for the detailed response and for raising your score. We truly appreciate the open-minded engagement from all reviewers. We answer your points in detail below, and kindly ask that you reassess the novelty and contribution of our paper with this context. Please note that space constraints prevented us from highlighting all challenges faced in this work. Our comments further address this issue, and we will include them in the final version.
> > >
> > > > Novelty remains the main weakness of the paper. The justification that MPNN is analogous to 1-WL like the proposed algorithm with KHC is not reasonable. When early MPNN models were proposed like GCN/GIN, it provided insights which were unknown during the time. Saying that neural extensions are needed for similar algorithms is not convincing by itself. I’m not able find new insights so far.
> > >
> > > We agree with the part that one should not align with any algorithm for the sake of it. Indeed, this is not our motivation: we take *inspiration* from KHC, to design the first *scalable, learnable, and complete* algorithm on planar graphs, with the *right inductive biases*. This is our fundamental contribution, which we substantiated theoretically and empirically — and further strengthened thanks to each reviewer’s input. We think this objective is ambitious and, to the best of our understanding, you agree with the *significance* of this objective.
> > >
> > > If our understanding is correct, your main objection lies in whether our approach is novel enough. The novelty discussion is always somewhat subjective, so we suggest a slight change in perspective and argue for the following:
> > > 1. PlanE closes an important gap in graph ML literature by achieving the earlier stated objective.
> > > 2. PlanE sets a standard for future graph ML research on planar graphs based on formal desiderata, and achieves these using modest computational resources.
> > >
> > > Based on (1)-(2), we strongly think that the graph ML community will benefit from PlanE and build on it to establish similar guarantees for their algorithm. Our work unlocks many new possible avenues for future work which could serve as a witness to its novelty in the long term.
> > >
> > > > From the KHC algorithm, it is clear that bi-connected and tri-connected components are the key to distinguishing planar graphs which was also shown in new experiments shared by authors. However, this is already known albeit independently of planar isomorphism [1]. I do not see additional novel insights we can get from the proposed approach.
> > >
> > > We are familiar with the work of Zhang et al., but we have difficulty understanding your precise argument. As you state, Zhang et al. do not study planar graphs. They also do not claim completeness over any graph class. Our study is therefore largely disjoint from Zhang el al. In fact, the only connection we can see is their study of biconnected components, i.e., they show that existing MPNNs (and most subgraph GNNs) cannot detect biconnectivity, with few exceptions, e.g., ESAN. This is important, because we include ESAN - a strong subgraph GNN baseline - in our analysis to better locate our contribution. It is clear that ESAN cannot achieve our desiderata (not scalable, no completeness result).
> > >
> > > Please note that our algorithm achieves much more than detecting components. In PlanE, components are carefully used in a specific way to ensure completeness over planar graphs, and this is highly non-trivial. It is very plausible for models to detect biconnectivity but still *remain incomplete* on planar graphs. We do not aim to detect substructures, or advertise a particular decomposition, but rather use these as a *means to an end*. Other means of achieving this goal are left for future research, and we hope this work inspires such endeavors.
> > >
> > > > Empirically, the model performs comparably to other SOTA methods, however is not better than them. There is benefit in the runtime being linear. However, as shown in [1], similar results can be achieved just by using biconnected components. Therefore, there is not much contribution in this regard.
> > >
> > > Our ablation study (Table 1 of *response.pdf*), empirically suggests the strength of a complete BasePlanE against its incomplete counterparts. The strong performance of BasePlanE against ESAN can be explained in the same way. Therefore, both theoretical and empirical evidence suggest the opposite: our results cannot be achieved by only using biconnected components. Theoretically, this is trivially true (i.e., considering a graph consisting of one tri-connected component), but interestingly, it is also prominent on real-world data. We therefore kindly ask you to take this into account.
> > >
> > > We hope this comment helps better convey the goals of our paper.

---

> > > > ### Comment · Reviewer_o9ac · 2023-08-21
> > > > **Reviewer response**
> > > >
> > > > I thank the authors for their additional response. However, I would like to keep my current score. I restate below the main reason for my rating.
> > > >
> > > > The authors have leveraged the KHC algorithm for planar graphs isomorphism which mainly depends on bi/tri-connected components. These components are pre-processed and incorporated into the message passing scheme to improve the expressive power.
> > > >
> > > > Although it is interesting, this does not shed further light which is already not known i.e. using bi-connected components (in this case tri-connected components as well) to improve expressivity.
> > > >
> > > > In my view, it would have been significantly novel if the proposed algorithm did not preprocess these components and could provably detect them and generate complete invariants on planar graphs.

---

> > > > > ### Author Response · Authors · 2023-08-21
> > > > >
> > > > > We thank the reviewer for their response. However, we find the following comment unsubstantiated:
> > > > >
> > > > > >Although it is interesting, this does not shed further light which is already not known i.e. using biconnected components (in this case triconnected components as well) to improve expressivity.
> > > > >
> > > > > A similar concern has been raised before and we have clarified that our approach does not amount to the implication of *“use biconnected or triconnected components”* and *“get more expressivity”*. While we appreciate the reviewer's feedback, this perspective is sadly undermining our work/effort for the following reasons:
> > > > >
> > > > > (1) As we stated in an earlier response, we do not solely provide increased expressive power, but *completeness guarantees*. This cannot be naively obtained by detecting biconnected or triconnected components.
> > > > >
> > > > > (2) As stated in our previous response, ESAN can detect bi-connectivity but how does this imply being complete on planar graphs? Similarly, it is known from classical literature that 2FWL can detect biconnectivity, but none of these algorithms are known to be complete on planar graphs. We would like to further restate that despite the lack of guarantees neither ESAN nor 2FWL-PPGN is nearly as scalable as BasePlanE as we have formally and empirically shown.
> > > > >
> > > > > (3) Our algorithm achieves much more than detecting components. In PlanE, components are carefully used in a specific way to ensure completeness over planar graphs, and this is highly non-trivial. It is very plausible for models to detect biconnectivity but still remain *incomplete on planar graphs*. We do not aim to detect substructures, or advertise a particular decomposition, but rather use these as a means to an end. Other means of achieving this goal are left for future research, and we hope this work inspires such endeavors.
> > > > >
> > > > > We are confident that our approach is the only one which achieves the stated desiderata in our paper: a scalable, complete, learnable algorithm on planar graphs with the right inductive biases. If there is a more efficient way of achieving this goal, this is clearly going to be challenging and subject of future study.

---

> > > > > > ### Comment · Reviewer_o9ac · 2023-08-21
> > > > > > **Reviewer response**
> > > > > >
> > > > > > Thanks for the response. Please find below my substantiation points.
> > > > > >
> > > > > > > As we stated in an earlier response, we do not solely provide increased expressive power, but completeness guarantees. This cannot be naively obtained by detecting biconnected or triconnected components.
> > > > > >
> > > > > > Completeness guarantee comes from aligning with KHC algorithm. In my view, this is not much insightful since aligning with a known complete algorithm on planar graphs naturally follows similar result in learning.
> > > > > >
> > > > > > > As stated in our previous response, ESAN can detect bi-connectivity but how does this imply being complete on planar graphs? Similarly, it is known from classical literature that 2FWL can detect biconnectivity, but none of these algorithms are known to be complete on planar graphs. We would like to further restate that despite the lack of guarantees neither ESAN nor 2FWL-PPGN is nearly as scalable as BasePlanE as we have formally and empirically shown.
> > > > > >
> > > > > > Biconnectivity itself is not complete. This is implied by KHC algorithm since it needs further tri-connected components. Therefore, I said we can get “similar” empirical results with bi-connectivilty and not “same”. Scalability is from KHC algorithm. There are no additional insights which are independent of alignement with the KHC algorithm. Nonetheless, I have previously improved my rating for scalability.
> > > > > >
> > > > > > > Our algorithm achieves much more than detecting components. In PlanE, components are carefully used in a specific way to ensure completeness over planar graphs, and this is highly non-trivial. It is very plausible for models to detect biconnectivity but still remain incomplete on planar graphs. We do not aim to detect substructures, or advertise a particular decomposition, but rather use these as a means to an end. Other means of achieving this goal are left for future research, and we hope this work inspires such endeavors.
> > > > > >
> > > > > > The presented algorithm does not even detect components. That is precisely the point. You preprocess them and then apply message passing with positional encodings on these components. Yes, models can detect Biconnectivity and remain not-complete because you need tri-connectivity as well for completeness. Do you have any counter-example graph to show that even if an algorithm can detect all cut-nodes, bi-connected and tri-connected components along with message passing and still remain not complete? If yes, please provide it and I’ll reevaluate.

---

> > > > > > > ### Author Response · Authors · 2023-08-21
> > > > > > >
> > > > > > > Thanks for your quick response:
> > > > > > > > Completeness guarantee comes from aligning with KHC algorithm. In my view, this is not much insightful since aligning with a known complete algorithm on planar graphs naturally follows similar result in learning.
> > > > > > >
> > > > > > > We would like to reiterate that the alignment with KHC itself is not a trivial process, and involved both an adaptation of components such as interleaving into neural-friendly modules, as well as re-deriving a new set of results distinctly from the KHC setup. Our results don't "naturally follow" KHC. They build on it, while introducing key inductive biases and carefully designed components, to achieve completeness. We urge the reviewer to consider this point seriously. It is not because we align with KHC that things simply fall into place.
> > > > > > >
> > > > > > > > Biconnectivity itself is not complete. This is implied by KHC algorithm since it needs further tri-connected components. Therefore, I said we can get “similar” empirical results with bi-connectivilty and not “same”. Scalability is from KHC algorithm. There are no additional insights which are independent of alignement with the KHC algorithm. Nonetheless, I have previously improved my rating for scalability.
> > > > > > >
> > > > > > > Detecting bi-connectivity is very different than completeness, as our later response will further establish. Nonetheless, we are very grateful for your engagement on scalability, and hope we can get our point across on this aspect as well.
> > > > > > >
> > > > > > > >The presented algorithm does not even detect components. That is precisely the point. You preprocess them and then apply message passing with positional encodings on these components. Yes, models can detect Biconnectivity and remain not-complete because you need tri-connectivity as well for completeness. Do you have any counter-example graph to show that even if an algorithm can detect all cut-nodes, bi-connected and tri-connected components along with message passing and still remain not complete? If yes, please provide it and I’ll reevaluate.
> > > > > > >
> > > > > > > Even if the algorithm detects triconnectivity and biconnectivity, this is not necessarily enough for distinguishing planar graphs; see the following cases for examples:
> > > > > > >
> > > > > > > (1) Consider the extreme case: the graph being a triconnected planar graph. In that case, there is a single triconnected component being the full graph, a single biconnected component (also the full graph) and no cut vertices (as the graph is also biconnected). Hence, detecting biconnected and triconnected components, as well as, cut nodes only provides trivial information, and we fall back to 1-WL.
> > > > > > >
> > > > > > > (2) For another example, consider the following: Take all 8 vertex, 3-regular planar graphs. There are exactly 3 such graphs, and all of them are biconnected. This means that there are no cut vertices, and due to the graphs being 3-regular, they are also trivially triconnected. Hence, simply detecting biconnectivity and triconnectivity will fall back to 1-WL, which cannot distinguish these graphs. Furthermore, the same argument can be applied to all graphs except for one in the P3R dataset (same construction, but with 10 nodes).
> > > > > > >
> > > > > > > We hope this answers your question.

---

> > > > > > > > ### Comment · Reviewer_o9ac · 2023-08-21
> > > > > > > > **Reviewer response**
> > > > > > > >
> > > > > > > > Thanks for the response.
> > > > > > > >
> > > > > > > > > We would like to reiterate that the alignment with KHC itself is not a trivial process, and involved both an adaptation of components such as interleaving into neural-friendly modules, as well as re-deriving a new set of results distinctly from the KHC setup. Our results don't "naturally follow" KHC. They build on it, while introducing key inductive biases and carefully designed components, to achieve completeness. We urge the reviewer to consider this point seriously. It is not because we align with KHC that things simply fall into place.
> > > > > > > >
> > > > > > > > Simply restating the same response multiple times does not prove anything. I have properly read your previous responses and replied there is no significant novel insights to be seen. If you would like to respond, please point out what exactly is the "non-trivial" part in designing your algorithm w.r.t KHC.
> > > > > > > >
> > > > > > > > > (2) For another example, consider the following: Take all 8 vertex, 3-regular planar graphs. There are exactly 3 such graphs, and all of them are biconnected. This means that there are no cut vertices, and due to the graphs being 3-regular, they are also trivially triconnected. Hence, simply detecting biconnectivity and triconnectivity will fall back to 1-WL, which cannot distinguish these graphs. Furthermore, the same argument can be applied to all graphs except for one in the P3R dataset (same construction, but with 10 nodes).
> > > > > > > >
> > > > > > > > Thanks for the examples. This will help evaluate novelty. Can you please point out what exactly is the component or algorithmic step which helps distinguish these 3 graphs - is it SPQR tree or Weinberg's code or something else? And is there any non-trivial contribution from your end on this part?

---

> > > > > > > > > ### Author Response · Authors · 2023-08-21
> > > > > > > > >
> > > > > > > > > >Simply restating the same response multiple times does not prove anything. I have properly read your previous responses and replied there is no significant novel insights to be seen. If you would like to respond, please point out what exactly is the "non-trivial" part in designing your algorithm w.r.t KHC.
> > > > > > > > >
> > > > > > > > > We actually did precisely this. In order to not repeat ourselves, we refer to our first response: the parts regarding interleaving, the node update equations, and learned component representations. If you consider these adaptations trivial then we unfortunately disagree.
> > > > > > > > >
> > > > > > > > > >Thanks for the examples. This will help evaluate novelty. Can you please point out what exactly is the component or algorithmic step which helps distinguish these 3 graphs - is it SPQR tree or Weinberg's code or something else? And is there any contribution from your end on this part?
> > > > > > > > >
> > > > > > > > > Thank you for acknowledging the counter-examples.  In this particular case, it is our implicit emulation of Weinberg codes in the BasePlanE architecture. Indeed, Weinberg codes are essential for the completeness result, but are not useful from a learnability perspective. Therefore, we propose a dedicated TriEnc component that implicitly can learn to emulate these codes (using a custom feed-forward MLP supplemented with positional encodings, etc.), while also leveraging node features. This in fact necessitated a proof on our side showing how our neural component can recover Weinberg codes. This result is an essential part of the overall proof we devise for the completeness of BasePlane.  Hence, we rely on learning additional structure (Weinberg code) similarly to KHC and establish a correspondence with the theory to obtain our overall result
> > > > > > > > >
> > > > > > > > > Please note that the reasons may vary depending on the examples and this response only considers the graphs under consideration.

---

### Official Review · Reviewer_hVMw · 2023-07-15

**Soundness:** 3 good
**Presentation:** 3 good
**Contribution:** 3 good
**Rating:** 6
**Confidence:** 3

**Summary:**

This work focuses on enhancing the representation power of GNNs in terms of distinguishing non-isomorphic graphs. Inspired by the classical planar graph isomorphism algorithm, the paper designs architectures within the proposed PLANE framework for learning complete invariants of planar graphs. The proposed framework achieves scalability while producing strong performance on planar graph benchmarks.

**Strengths:**

1. In contrast to previous approaches that cannot strike a balance between algorithmic efficiency and representation power, the proposed framework, PLANE, can efficiently learn isomorphism-complete invariant functions for planar graphs.

2. The architectural designs within the PLANE framework draw inspiration from classic planar graph isomorphism algorithms and can provably distinguish between any pair of non-isomorphic planar graphs.

3. The efficacy of the proposed framework is validated through extensive experiments conducted on both synthetic datasets and real-world benchmarks, further highlighting its effectiveness in practical scenarios.

**Weaknesses:**

1. While the synthetic datasets used in the study provide some insights into the theoretical power of the proposed framework, they may not fully capture its potential. Notably, the clustering coefficient and EXP datasets can be easily handled by a class of models known as subgraph GNNs ([1, 2], and [7] in the main paper), which can be efficient enough [3]. To showcase the representation power of the proposed framework, it is recommended to include additional datasets, such as the strongly regular graphs used in CWN ([9] in the main paper) or generate planar graph datasets.

2. Furthermore, it is important to report the runtime of the proposed framework, including the preparation time (e.g., time to generate the BlockCUT or SPQR trees) and the overall runtime encompassing training and inference. Comparisons should be made with existing models, including the classic MPNN and baseline models like ESAN, to demonstrate the efficiency of the proposed framework.

[1] You J, Gomes-Selman J M, Ying R, et al. Identity-aware graph neural networks. AAAI 2021.

[2] Zhang M, Li P. Nested graph neural networks. NeurIPS 2021.

[3] Zhao L, Jin W, Akoglu L, et al. From Stars to Subgraphs: Uplifting Any GNN with Local Structure Awareness. ICLR 2022.


**Questions:**

See Weakness.

---

> ### Author Rebuttal · Authors · 2023-08-09
>
> Thank you for your constructive feedback. We respond to your points below:
>
> > While the synthetic datasets used in the study provide some insights into the theoretical power of the proposed framework, they may not fully capture its potential. Notably, the clustering coefficient and EXP datasets can be easily handled by a class of models known as subgraph GNNs ([1, 2], and [7] in the main paper), which can be efficient enough [3]. To showcase the representation power of the proposed framework, it is recommended to include additional datasets, such as the strongly regular graphs used in CWN ([9] in the main paper) or generate planar graph datasets.
>
> Thanks for this suggestion! We agree that running our model on larger (synthetic) graphs would further demonstrate the benefits of our specialized approach. Please note that we sought to use strongly regular graphs as in CWN, but there are only 7 strongly regular planar graphs, which are in fact $\text{K}_1$, $\text{K}_2$, $\text{K}_3$, $\text{K}_4$, $\text{C}_4$, $\text{C}_5$, and $\overline{3\text{K}_2}$. However, these graphs are easily distinguishable by standard GNNs, as all but $\text{C}_4$ and $\text{K}_4$ have a different number of nodes, and the latter pair have a different number of edges, rendering the whole task trivial.
>
> As a result, we designed a new synthetic dataset based on 3-regular planar graphs, and experimented with BasePlanE, GIN and PPGNs. In this experiment, we observed that GIN struggles to go beyond a random guess, whereas BasePlane and PPGNs perfectly solve the problem, achieving 100\% accuracy. We provide the full experimental setup and details in the global response, and our results can be found in Table 2 of *response.pdf*. We hope that the inclusion of this new experiment addresses your concern.
>
> > Furthermore, it is important to report the runtime of the proposed framework, including the preparation time (e.g., time to generate the BlockCUT or SPQR trees) and the overall runtime encompassing training and inference. Comparisons should be made with existing models, including the classic MPNN and baseline models like ESAN, to demonstrate the efficiency of the proposed framework.
>
> Thank you! We agree it is important to illustrate this, and following your feedback, we conduct a runtime experiment on real-world planar graphs. We compare the wall-clock time of BasePlanE and 2-FWL-expressive PPGN. We observed that both the pre-processing and inference of BasePlanE run efficiently and scale very well, whereas PPGN quickly fails to run for larger graphs. This highlights that PlanE is a highly efficient option for inference on large-scale graphs compared to higher-order alternatives. We report the full runtime results in Table 3 of  *response.pdf*, and describe our findings in detail in the global response.
>
> Please note that the paper includes a dedicated complexity analysis in Appendix A, which stated the asymptotic efficiency of BasePlanE. The runtime of BasePlanE is $O(|V| d^2)$ with a (one-off) preprocessing time $O(|V|^2)$. In practical terms, this makes BasePlanE linear in the number of graph nodes after preprocessing. This is very scalable as opposed to 3-FWL (or 4-WL) which requires about $O(|V|^4 \log |V|)$ steps to reach a stable coloring. We added this explanation to the main paper based on your feedback.

---

> > ### Comment · Reviewer_hVMw · 2023-08-14
> >
> > Thanks for your response. In Table 3 in the submitted "response.pdf", what is the running time of MPNNs (e.g., GIN) and subgraph GNNs (ESAN)?

---

> > > ### Author Response · Authors · 2023-08-14
> > >
> > > We ran BasePlanE against 2FWL-expressive PPGN to show that BasePlanE is much more scalable than its higher-order alternatives.  We expect GIN to be fastests given its simplicity and $O(|E|)$ runtime. As for ESAN, the runtime depends on many factors, but it is likely to be slower than BasePlanE for training, since according to the original paper, it scales with $O($#subgraphs $* |V| *$ max node degree), where #subgraphs scales with $|V|$. To put things in better perspective, we will also run GIN and ESAN and include their runtime in Table 3. Please note that we prioritised PPGN during the rebuttal period due to time constraints, because this comparison appeared as the most essential one (by the expressiveness perspective). That being said, we would like to reiterate that BasePlanE is the only architecture which is provably complete on planar graphs among these architectures.

---

> > > > ### Comment · Reviewer_hVMw · 2023-08-14
> > > >
> > > > Thanks for your additional response and effort, and good luck with your submission!

---

> > > > > ### Author Response · Authors · 2023-08-14
> > > > >
> > > > > Thanks for all the constructive feedback and for revising your score! We will include all the findings from the rebuttal/discussion period into the new version of the paper.

---

> > > > > > ### Author Response · Authors · 2023-08-15
> > > > > >
> > > > > > To follow up on this comment, we now have the runtime results also for GIN and ESAN on the large planar graphs based on TIGER Alaska. The training runtimes for all models are as follows:
> > > > > >
> > > > > > | Dataset    | GIN | BasePlane (Train) | PPGN (Train)| ESAN (Train)|
> > > > > > | -------- | ------- | ------- |------- | ------- |
> > > > > > | TIGER-Alaska-2K  | 0.007 sec | 0.1  sec |5.9 sec| 87.73 sec|
> > > > > > | TIGER-Alaska-10K | 0.02 sec  | 0.33  sec | OOM | OOM|
> > > > > > | TIGER-Alaska-93K    | 0.18 sec  | 2.2 sec |OOM |OOM|
> > > > > >
> > > > > > The preprocessing times are as reported in Table 3 and the preprocessing time for ESAN on TIGER-Alaska-2K is 4.6 sec. This confirms our expectation: GIN is fastest and BasePlanE is the only other model which can scale up to all datasets. Indeed, both ESAN and PPGN fail on TIGER-Alaska-10K and TIGER-Alaska-93K. Somewhat surprisingly, ESAN training time is slower than PPGN training time on TIGER-Alaska-2K which is likely due to the fact that the max degree of TIGER-Alaska-2K is 620, which the ESAN algorithm depends on. We will incorporate these findings into Table 3 in the next version of the paper. Thank you once again for the suggestion!

---

### Official Review · Reviewer_fXaJ · 2023-07-17

**Soundness:** 3 good
**Presentation:** 1 poor
**Contribution:** 3 good
**Rating:** 6
**Confidence:** 3

**Summary:**

This paper improves the graph isonophism inspired GNN design for a particular type of graph, planar graphs.

It utilizes KHC algorithm to generate a symbolic code for each graph. Follow the sequence, they apply GNN to recursively get the whole graph representation, which shall have the good property to be invariant.

The results show good performance than some prior baselines such as GIN.

**Strengths:**

1. The paper levereages existing algorithm for isomorphism testing of planar graph, which is both effective and efficient;

2. The authors prove that there exist a parametrization of GNN followed their message passing order that could distinguish any two planar graphs.

3. Empirical results on chemical graphs show their advantage for real-world graphs.

**Weaknesses:**

My major concern for this papaer is that the presentation makes it really hard to understand the algorithm.

In sec 4 you define CODE, but I didn't see where you utilize these codes in the algorithm? Seem that all you utilize is the sequence of walk and the results of SPQR. It's very hard for me to understand what each notation is referring to (is it a node or a subgraph or a sequence). Also I don't see the definition of X.

I highly suggest the authors improve the sec 4 & 5 to make it easier to understand what you did in this algorithm. It would be better to add a pseudocode for illustration.

Also, as you claim the achieve efficient calculation, it's better to show the time & memory usage for you to train & infer some graph datasets, especially over some larger graphs.

**Questions:**

The authors mention the message passing and the KHC algorithm could be run in parallel, but I guess it means when you do GPU calculation you could do KHC on CPU for the next batch? Could the authors elaborate this part?

**Limitations:**

THe authors include a limitation statement,.

---

> ### Author Rebuttal · Authors · 2023-08-09
>
> Thank you for your constructive feedback. We respond to your main points below:
>
> > My major concern for this papaer is that the presentation makes it really hard to understand the algorithm.
>
> Thanks for raising this point. We have now included a figure visualising the full decomposition and encoding process. This is Figure 1 and can be found in *response.pdf*.  We have also added discussions on complexity analysis as well as experimental setup in the main paper. Please refer to the global response for more information.
>
> > In sec 4 you define CODE, but I didn't see where you utilize these codes in the algorithm? Seem that all you utilize is the sequence of walk and the results of SPQR.
>
> Yes, we do not directly use CODEs as part of our model, but instead build on CODE to prove completeness results. In our proofs, we show how our model can yield embeddings which are in bijection with the corresponding codes of the formal Weinberg algorithm. This is similar to the GNN expressiveness results: every (maximally expressive) GNN layer aligns with a step of 1-WL algorithm if the computed node embeddings refine the 1-WL hash and vice versa. Therefore, CODEs can be seen as a meaningful abstraction through which alignment with the classical planarity algorithm can be explained.
>
> > It's very hard for me to understand what each notation is referring to (is it a node or a subgraph or a sequence).
>
> We will clarify all these definitions in the paper. Please let us know if there are any particular aspects of our notation you would like us to address.
>
> >  Also I don't see the definition of X.
>
> We define this on page 3: $\chi(u)$ denotes the set of the children of node $u$.
>
> > I highly suggest the authors improve the sec 4 & 5 to make it easier to understand what you did in this algorithm. It would be better to add a pseudocode for illustration.
>
> Thanks for the suggestion! We have added an illustrative figure visualizing the steps and intermediate outputs of our model, which we hope will improve these sections. This figure can be found as Figure 1 in our *response.pdf*. We will also provide more information on runtime, component designs, and experimental setup in the main paper. We considered adding pseudo-code as per your suggestion, but found this to be harder to follow than the overall figure.
>
> > Also, as you claim the achieve efficient calculation, it's better to show the time & memory usage for you to train & infer some graph datasets, especially over some larger graphs.
>
> Thank you! To validate this empirically, we conduct a runtime experiment on real-world planar graphs. We compare the wall-clock time of BasePlanE and 2-FWL-expressive PPGN. We observed that both the pre-processing and inference of BasePlanE run efficiently and scale very well, whereas PPGN quickly fails to run for larger graphs. This highlights that PlanE is a highly efficient option for inference on large-scale graphs compared to higher-order alternatives. We report the full runtime results in Table 3 of  *response.pdf*, and describe our findings in detail in the global response.
>
> Please also note that the paper includes a dedicated complexity analysis in Appendix A, which stated the asymptotic efficiency of BasePlanE. The runtime of BasePlanE is $O(|V|  d^2)$ with a (one-off) pre-processing time $O(|V|^2)$. In practical terms, this makes BasePlanE linear in the number of graph nodes after pre-processing. This is very scalable as opposed to 3-FWL (or 4-WL) which requires about $O(|V|^4 \log |V|)$ steps to reach a stable coloring. We added this explanation to the main paper.
>
> > The authors mention the message passing and the KHC algorithm could be run in parallel, but I guess it means when you do GPU calculation you could do KHC on CPU for the next batch? Could the authors elaborate this part?
>
> Thank you for the question. We assume you refer to the statement on lines 203-204 (page 5). We meant the following with this statement: there is a parallel in the *ideas* between the component code generations of KHC, and the encoders of PlanE.
>
> Coincidentally, however, paralel computability is a very good point: it is indeed possible to parallelize the batches in the way you point out. Furthermore, it could be possible to parallelize the KHC pre-processing for a single graph instance, by evaluating the lexicographically smallest TriCode in parallel (which is the quadratic bottleneck in the KHC algorithm). We will integrate this in the discussion of the runtime.

---

> > ### Comment · Reviewer_fXaJ · 2023-08-12
> > **Response**
> >
> > Thanks for authors responding my questions.
> >
> > Fig 1 looks nice, Table 3 as well as the complexity also are very helpful. Please try to add them into the paper.
> >
> > I'll raise my score to weak accept. I do think this paper has great value if the presentation could be improved such that readers could more easily find which part is proof and which part is algorithm (and how they could be implemented). Looking forward to seeing the updated (improved) version.

---

> > > ### Author Response · Authors · 2023-08-12
> > >
> > > Thank you for going through the response: we are glad you find great value in the paper and we thank you for raising your score. We will carefully improve the presentation of the paper and integrate all the feedback and the findings summarised in the rebuttal into the updated version.

---

### Official Review · Reviewer_fYDw · 2023-07-18

**Soundness:** 3 good
**Presentation:** 3 good
**Contribution:** 3 good
**Rating:** 6
**Confidence:** 4

**Summary:**

Message passing based neural network is well-known for its limited expressivity of identifying graphs, while higher order neural network like  IGNs and SetGNNs are not scalable. Instead of trying to solve complete invariant for any graphs, this paper focuses on finding complete invariant for planar graphs. The author revisited the literature of the complete graph isomorphism test for planar graphs, which does hierarchical graph encoding based on block cut tree and SPQR tree (that decomposes a graph into k-connected components iteratively with k=1,2,3). Furthermore, the author proposes an neural network variant that aligns with the planar graph isomorphism test procedure, and proved that it is theoretically compete invariant for planar graphs. The author shows the advantage of these methods over simulation datasets as well as real-world datasets.

**Strengths:**

1. Given planar graphs are widely appeared in real-world (road networks, circuits, moleculers and so on), designing an efficient graph neural network while still be expressive enough is an important problem (Powerful than 3-FWL but more efficient than ). The author makes a first step by revisiting planar graph isomorphism test. The contribution and its foundation is solid.

2. As planar graph isomorphism test makes use of hierachical decomposition in a top-down manner and then encode each components in a bottom-up manner, the author's proposed neural network is kind of bottom-up hierachical graph encoding. The direction of hierarchical graph representation learning is promising and deserves more attention, although being a hard direction.

3. The designing of neural network architecture clearly follows the procedure, is well supported by planar graph isomorphism test theory.

**Weaknesses:**

1. The presentation is not clear enough, the author better provide a good figure to illustrate the hierachical steps clearly to help the reader understand the main idea in a minute.

2. Some unclear notations, such as line 232, tilde h_u^(l).

3. Very importantly, the paper lacks the comprehensive complexity analysis, giving the goal of this paper is designing a powerful but also efficient method comparing with 3-FWL. Also, the author should discuss the impact of using the procedure of graph decomposition inside planar graph isomorphism test (the top-down step). First, the top-down step is kind of preprocessing that needs for the designed neural network which may introduces additional runtime and computational complexity. Second, as the top-down step is highly aligned with planar graph, it doesn't support other type of graphs.

4. Although the design of neural network tries to follow the planar graph isomorphism test closely, the author should discuss and do some ablation study over these designs.

5. Experimental step is too vague and unclear. For all experiments, the description of experimental setup is incomplete. The hyperparameter is set completely with a single number and no discussion of hyperparameter tuning for different methods. It is unfair for baselines to use the same hyperparameter as the proposed method. The author should do hyperparameter tuning for all methods properly.

5. Baselines are not SOTA. Only ESAN is used, while other recent baselines like SetGNN (Lingxiao Zhao, et al.) and SSWL (Bohang Zhang, et al. 2023) should be considered. Also I strongly recommend the author to tune hyperparameter in a systematic way.

**Questions:**

I listed all suggestions and questions in weakness.
While this paper is interesting, I hope some important problems can be improved during the rebuttal period.

**Limitations:**

The main limitation is that the designed architecture relies on the fixed top-down graph decomposition method for planar graph. The designed architecture should at least to be able to run all graphs while being incomplete graph invariant.

---

> ### Author Rebuttal · Authors · 2023-08-09
>
> We thank the reviewer for their constructive feedback, and respond to their main points below:
>
> > The presentation is not clear enough, the author better provide a good figure to illustrate the hierachical steps clearly to help the reader understand the main idea in a minute.
>
> Following your suggestion, we added a figure visualising the PlanE pipeline, which can be found as Figure 1 in our response file. This figure shows the graph decomposition and the corresponding encoding steps. Please let us know if you have any further suggestions, and we will do our best to accommodate these.
>
> > Some unclear notations, such as line 232, tilde h_u^(l).
>
> This is indeed a typo. We corrected this to $\mathbf{h}_{\gamma_u}^{(\ell)}$.
>
> > Very importantly, the paper lacks the comprehensive complexity analysis, giving the goal of this paper is designing a powerful but also efficient method comparing with 3-FWL.
>
> Please note that the paper includes a dedicated complexity analysis in Appendix A, which grounds the asymptotic efficiency of BasePlanE. The runtime of BasePlanE is $O(|V| d^2)$ with a (one-off) pre-processing time $O(|V|^2)$. In practical terms, this makes BasePlanE linear in the number of graph nodes after preprocessing. This is very scalable as opposed to 3-FWL (or 4-WL) which requires about $O(|V|^4  \log |V|)$ steps to reach a stable coloring. Following your feedback, we added this explanation to the main paper.
>
> To validate this empirically, we conduct a runtime experiment on real-world planar graphs. We compare the wallclock time of BasePlanE and 2-FWL-expressive PPGN. We observed that both the pre-processing and inference of BasePlanE run efficiently and scale very well, whereas PPGN quickly fails to run for larger graphs. This highlights that PlanE is a highly efficient option for inference on large-scale graphs compared to higher-order alternatives. We report the full runtime results in Table 3 of  *response.pdf*, and describe our findings in detail in the global response.
>
> > Also, the author should discuss the impact of using the procedure of graph decomposition inside planar graph isomorphism test (the top-down step). First, the top-down step is kind of preprocessing that needs for the designed neural network which may introduces additional runtime and computational complexity.
>
> We discuss the runtime complexity of the decomposition step in Appendix A. In essence, we emulate the classical Weinberg algorithm in our pre-processing to obtain all necessary components, and this incurs a *one-time, worst-case* pre-processing cost of $O(|V|^2)$. This worst-case applies only when the *entire graph is a single tri-connected component*, which is rare in practice.
>
> > Second, as the top-down step is highly aligned with planar graph, it doesn't support other type of graphs.
>
> Yes: BasePlanE is explicitly aligned with KHC, and does not immediately apply beyond planar graphs as stated in our limitations section. However, our main contribution is to develop a specialized, complete, and learnable network to efficiently learn functions over planar graphs. While our work *can* be applied to other graphs with simple modifications, this will either come at the expense of (i) completeness or (ii) efficiency. If we forgo the completeness criteria (as it is the case for existing GNNs), then PlanE can easily be extended to general graphs by choosing appropriate encoders for TriEnc and BiEnc. The resulting model will not be complete but still very expressive, since most existing models cannot even detect biconnectivity; see, e.g., Zhang et al. (2023).
>
> The study of graph components has been very influential in graph theory, but remains much less explored in the context of graph ML. Hence, we think our work paves the way for designing powerful architectures, which additionally exploit different graph components.
>
> > Although the design of neural network tries to follow the planar graph isomorphism test closely, the author should discuss and do some ablation study over these designs.
>
> Thanks for this! We experimented with several simplifications of BasePlanE on ZINC. In summary, we observed  that BasePlanE substantially benefits from each of the components in its message passing update. Indeed, dropping any individual component leads to loss of performance. Moreover, the combination of standard message passing with KHC component decompositions proved essential to our results. Hence, each component is ultimately beneficial within the overall PlanE framework. We provide a comprehensive discussion on this experiment in our global response, and the full results can be found in Table 1 of *response.pdf*
>
> > Experimental step is too vague and unclear. For all experiments, the description of experimental setup is incomplete. The hyperparameter is set completely with a single number and no discussion of hyperparameter tuning for different methods. It is unfair for baselines to use the same hyperparameter as the proposed method. The author should do hyperparameter tuning for all methods properly.
>
> All details of our experimental setup are reported in the appendix.  Based on your feedback, we included these details into the main body.  We use exactly the same hyperparameter search protocols from the literature for each dataset for fairness with all models. This is actually constraining our own model’s tuning, as opposed to the other way around.
>
> > Baselines are not SOTA. Only ESAN is used, while other recent baselines like SetGNN (Lingxiao Zhao, et al.) and SSWL (Bohang Zhang, et al. 2023) should be considered.
>
> Thanks for bringing these to our attention! We are happy to include these in the next version of our work.
>
> > Also I strongly recommend the author to tune hyperparameter in a systematic way.
>
> We understand your concern, but we were systematic with the experimental setup, as detailed in our response.

---

> > ### Comment · Reviewer_fYDw · 2023-08-11
> > **Response**
> >
> > Thank you for additional work. I like the figure for visualization. Altough it still has some limitations, I believe the method has good inspiration for the community towards bringing more theoretical graph algorithms to GNN area. I revise the score accordingly.

---

> > > ### Author Response · Authors · 2023-08-12
> > >
> > > Thank you for going through our response and revising your score.

---

### Author Rebuttal · Authors · 2023-08-09

We thank the reviewers for their comments. We have responded to each concern in detail in our individual responses. In this global response, we include a response file, *response.pdf*, containing the results of additional experiments for your reference.
The changes made during the rebuttal can be summarized as follows:

1. **Overall figure (Reviewers fYDw, fXaJ):**
We provide a figure visualizing the PlanE pipeline in our response file, which details all the steps involved in our model’s computation. This figure shows the pre-processing steps, namely computing the block-cut and SPQR trees, and also illustrates the mapping from these to embeddings at all the different levels of our model, leading up to the final node-level representation update. We hope this figure provides a much clearer picture of our contribution.
2. **Runtime analysis (Reviewers fYDw, hVMw, o9ac):**

    * Theoretically, the runtime of BasePlanE is $O(|V| d^2)$ with a (one-off) pre-processing time $O(|V|^2)$. In practical terms, this makes BasePlanE linear in the number of graph nodes after preprocessing. This is very scalable as opposed to 3-FWL (or 4-WL) which requires about $O(|V|^4 \log |V|)$.

    * To validate this empirically, we conduct a runtime experiment on real-world planar graphs based on the geographic faces of Alaska from the dataset TIGER, provided by the U.S. Census Bureau. The original dataset is TIGER-Alaska-93K and has 93366 nodes. We also extract the smaller datasets TIGER-Alaska-2K and TIGER-Alaska-10K, which are subsets of the original dataset with 2000 and 10000 nodes, respectively. We compare the wallclock time of BasePlanE and 2-FWL-expressive PPGN. We observed that both the pre-processing and inference of BasePlanE run efficiently and scale very well, whereas PPGN quickly fails to run for larger graphs. This highlights that PlanE is a highly efficient option for inference on large-scale graphs compared to higher-order alternatives. We report the full runtime results in Table 3 of  *response.pdf*.

3. **New expressiveness experiment (Reviewers fYDw, hVMw, o9ac):**
We propose a new synthetic dataset based on 3-regular planar graphs, and experiment with BasePlanE, GIN and 2-FWL-expressive PPGN. For this experiment, we generated all 3-regular planar graphs of size 10, leading to exactly 9 non-isomorphic graphs. For each such graph, we generated 50 isomorphic graphs by permuting their nodes. The task is then to predict the correct class of an input graph, where the random accuracy is $\sim$11\%. We report accuracy results in Table 2 of our response file. As expected, GIN struggles to go beyond a random guess, whereas BasePlane and PPGNs easily solve the task, achieving 100\% accuracy. Note that PPGNs can solve this task because these graphs are 2-FWL-distinguishable.

4. **Ablation experiments (Reviewers fYDw, o9ac):**
We additionally conduct extensive ablation studies on each component in our model update equation using the ZINC 12k dataset. We report the final results in Table 1 of *response.pdf* and summarize them here:
    * *BasePlanE (no readout):* We removed the global readout term from the update formula and MAE worsened by 0.003.
    * *BasePlanE (no CutEnc):* We removed the cut node term from the update formula and MAE worsened by 0.003.
    * *BasePlanE (no neighbors):* We additionally removed the neighbor aggregation from the update formula and MAE worsened by 0.023.
    * *BasePlanE (only BiEnc):* We only used the triconnected components for the update formula and MAE worsened by 0.021.
    * *BasePlanE (only TriEnc):* We only used the triconnected components for the update formula and MAE worsened by 0.016.

    Hence, BasePlanE significantly benefits from each of its components: the combination of standard message passing with component decompositions is essential to obtaining our reported results.

We hope that our answers address your concerns along with the new experiments. We are looking forward to a fruitful discussion period.

---

### Decision · Program_Chairs · 2023-09-21

**Decision:**

Accept (poster)

**Comment:**

This paper focuses on the family of planar graphs and proposes a new GNN inspired by the classical planar graph isomorphism algorithm of Hopcroft and Tarjan. The proposed learning algorithm extends beyond the given algorithm while maintains the theoretical guarantee. Experiments on synthetic and real-world datasets validates the proposed solution. Despite some concerns regarding the novelty as the method is inspired by the classical algorithm, we still consider the contribution significant enough; it extends beyond the original algorithm by learning a graph representation that can be used broadly for different learning tasks.